# Characterizing and engineering post-translational modifications with high-throughput cell-free expression

Derek A. Wong [1,2,3,16], Zachary M. Shaver[2,3,4,5,16], Maria D. Cabezas[1,2,3], Martin Daniel-Ivad [6,7], Katherine F. Warfel[1,2,3], Deepali V. Prasanna[1,2,3], Sarah E. Sobol[1,2,3], Regina Fernandez [1,2,3], Fernando Tobias [8,9], Szymon K. Filip[10], Sophia W. Hulbert[11], Peter Faull [10], Robert Nicol[6], Matthew P. DeLisa[11,12,13], Emily P. Balskus [6,7,14] ✉, Ashty S. Karim [1,2,3] ✉ & Michael C. Jewett [1,2,3,15] ✉

Post-translational modifications (PTMs) are important for the stability and function of many therapeutic proteins and peptides. Current methods for studying and engineering PTMs are often limited by low-throughput experimental techniques. Here we describe a generalizable, in vitro workflow coupling cell-free gene expression (CFE) with AlphaLISA for the rapid expression and testing of PTM installing proteins. We apply our workflow to two representative classes of peptide and protein therapeutics: ribosomally synthesized and post-translationally modified peptides (RiPPs) and glycoproteins. First, we demonstrate how our workflow can be used to characterize the binding activity of RiPP recognition elements, an important first step in RiPP biosynthesis, and be integrated into a biodiscovery pipeline for computationally predicted RiPP products. Then, we adapt our workflow to study and engineer oligosaccharyltransferases (OSTs) involved in protein glycan coupling technology, leading to the identification of mutant OSTs and sites within a model vaccine carrier protein that enable high efficiency production of glycosylated proteins. We expect that our workflow will accelerate design-build-test-learn cycles for engineering PTMs.

Protein- and peptide-based biologics play an important role in treating and preventing a wide variety of illnesses. Currently, about 30% of all new US Food and Drug Administration (FDA) approved therapeutics entering the clinical setting are protein biologics[1]. Common protein-based therapeutics include antibodies[2], blood coagulants[3,4], and vaccines[5,6], among others[7]. Peptide drugs continue to mature as important options for treating microbial infection[8] and diabetes[9], among other conditions[10], with over 800 new peptide therapeutics either in clinical development or undergoing preclinical studies[11]. Understanding how to design and produce protein- and peptide-based

therapeutics with optimal characteristics continues to be a major focus in biological research.

For many biologics, post-translational modifications (PTMs) are important for stability and activity. Examples of PTMs include cyclization[12], methylation[13], β-hydroxylation[14], glycosylation[15], and sulfation[16], among many others[17]. Unfortunately, workflows for studying PTMs are often low throughput. For example, studies screening libraries of PTM installing enzymes or protein substrates often require overexpression of each variant in individual strains and labor-intensive protein purification steps. These methods are then coupled with

low-throughput analytical methods such as mass spectrometry[18–20], Western blotting[21], or ELISA[22], which are often time intensive or involve complex data analysis. Additionally, techniques used to directly measure interactions between PTM installing enzymes and their substrates, such as fluorescence polarization[23], co-crystallization of the substrate in the enzyme active site[24,25], and isothermal titration calorimetry (ITC)[26], often limit studies to single digits or tens of variants.

Advances in cell-free gene expression (CFE) systems[27] have enabled the parallelized expression of proteins and peptides in hours, which can facilitate the rapid characterization and engineering of PTMs. CFE systems[28–30] use transcription and translation machinery, rather than living cells, supplemented with additional cofactors, energy sources, salts, and a DNA template to produce a desired protein. CFE systems have been successfully applied to a variety of high-throughput bioengineering applications, such as engineering transcription factors[31,32], constructing metabolic[33–36] and glycosylation pathways[37–39], and studying the substrate promiscuity of various PTM installing enzymes[40–42]. However, many of these applications rely on liquid chromatography and mass spectrometry-based approaches or the ability to connect the targeted protein function with a visual output such as superfolder green fluorescent protein (sfGFP) production. Matching the throughput of CFE, AlphaLISA[43] is an in-solution, bead-based assay version of ELISA that is amenable to acoustic liquid handling robots and small (1–2 µL) reaction sizes in 384- or 1,536-well plate formats and has previously been used with cell-free systems to assess protein-protein interactions[44–46]. By requiring only liquid transfer and incubation steps, AlphaLISA facilitates the analysis of hundreds to thousands of reactions in hours.

Here, we describe a general in vitro, plate-based platform for characterizing and engineering PTMs using CFE and AlphaLISA which we apply to both (i) ribosomally synthesized and post-translationally modified peptides (RiPPs) and (ii) glycoproteins. To begin, we show that our workflow can be used to detect interactions between RiPP recognition elements (RREs) and their native precursor peptides, a key first step in the biosynthesis of many RiPP products[47]. We then characterize peptide residues important for RRE binding and assess RRE binding of computationally predicted RiPP products. By modifying the CFE portion of the workflow, we then directly measure the enzymatic attachment of glycans onto proteins. From a library of 285 unique enzyme variants, we identify 7 high-performing mutants, including a single mutant with a 1.7-fold improvement of glycosylation with a clinically relevant glycan. Finally, we systematically characterize accessible sites within an FDA approved carrier protein for protein glycosylation. We expect that our workflow will accelerate the characterization and engineering of PTMs important for protein- and peptide-based therapeutics.

## Results

### A cell-free AlphaLISA-based workflow can detect RRE-peptide interactions

The goal of our work was to develop a robust, high-throughput, and generalizable workflow that expedites the ability to characterize and engineer PTMs on peptides and proteins. Key to this development was the optimized integration of CFE and AlphaLISA, as well as the ability to study different classes of PTMs.

We chose to first apply our workflow to RiPPs (e.g., lanthipeptides[48,49], thiopeptides[50–52]) due to growing interest in their use as antimicrobial therapeutics[53–59]. While mature RiPPs vary in amino acid composition, RiPPs originate as a precursor peptide typically composed of an N-terminal leader sequence and C-terminal core sequence[60]. Tailoring enzymes encoded within the same biosynthetic gene cluster (BGC) as the precursor peptide recognize a portion of the leader sequence and install PTMs on the core

sequence, producing the mature RiPP[60]. In around 65% of RiPP classes produced in prokaryotes, the recognition of the leader sequence by tailoring enzymes is facilitated by a standalone protein or portion of a fusion protein containing a RiPP precursor peptide recognition element (RRE)[47,61]. In the absence of the RRE, individual reactions catalyzed by the tailoring enzymes often suffer from slow kinetics and low conversion rates[62]. Yet, despite their importance in catalyzing RiPP formation, current methods for studying interactions between RREs and their peptide substrate are low-throughput (e.g. fluorescence polarization[61,63,64] and co-crystallization[24]).

To begin, we selected a panel of 13 RREs from a range of RiPP classes. We initially assessed their expression in PURE*frex* via incorporation of FluoroTect™ Green$_{Lys}$ fluorescently labeled lysine (Supplementary Fig. 1). For 9 of these proteins, we tested the native sequence as well as fusion proteins in which the predicted RRE domain was fused to maltose-binding protein (MBP) due to their size and/or origin from a radical *S*-adenosyl-L-methionine (SAM) enzyme that could potentially make expression difficult. While some of the full-length constructs did produce soluble protein, we generally saw better expression when constructs were fused to MBP. We next tested the functionality of these MBP-tagged RRE proteins in an AlphaLISA assay with each of their respective peptide substrates (Fig. 1a). To do so, we expressed RRE fusion proteins and N-terminally sFLAG-tagged peptide substrates in individual PURE*frex* reactions. We then assayed for RRE-peptide recognition by mixing an RRE protein-expressing PURE*frex* reaction and the corresponding peptide substrate-expressing reaction with anti-FLAG donor beads and anti-MBP acceptor beads. Only in instances in which the RRE binds the peptide will the acceptor and donor bead be brought within close enough proximity to produce a chemiluminescent signal. A cross-titration of four different RRE-peptide pairs (PqqD, TbiB1, HcaF, TbtF) across multiple dilutions revealed a clear binding pattern consistent with RRE-peptide engagement (Fig. 1b–e), which we do not observe when assaying MBP only with the respective peptides (Supplementary Fig. 2).

### Mapping RRE-peptide binding landscapes informs design

We next asked whether we could characterize a peptide-binding landscape to inform the design of a synthetic peptide capable of binding to a naturally occurring RRE. To do this, we chose the RRE domain of TbtF, the cyclodehydratase involved in thiomuracin[65–68] biosynthesis, and its leader sequence of TbtA (Fig. 2a). Mutating residues L(-32), L(−29), M(-27), D(-26), and F(-24) within the leader sequence of TbtA to an alanine was previously shown using fluorescence polarization to reduce binding affinity to TbtF[66]. By creating an alanine positional scanning library, we demonstrated that our method could achieve similar results to those using fluorescence polarization as evidenced by a >100-fold decrease in AlphaLISA signal compared to the wild-type peptide sequence for all noted mutations. We also found that the mutation D(-30)A resulted in a >100-fold decrease in AlphaLISA signal. By using CFE combined with AlphaLISA, we characterized the peptide-binding landscape within hours without conventional cloning, transformation, expression, and purification workflows normally required for fluorescence polarization competition assays.

We then used TbtA's peptide-binding landscape to design a synthetic peptide capable of binding to TbtF. We started with a synthetic peptide sequence the same length as the leader sequence of TbtA that does not bind to TbtF (Fig. 2b; peptide variant 2), using the first 40 amino acids of sfGFP with a G(-18)T mutation to ensure all residues in the region of interest differed from the wild-type TbtA leader sequence. We then created peptide variants by replacing residues in the synthetic peptide with residues identified from the alanine scan as important for binding by TbtF, starting with the six residues (L(-32), D(-30), L(-29), M(-27), D(-26), and F(-24)) that when mutated to an alanine resulted in the greatest decrease in AlphaLISA signal. We were unable to detect binding interactions between this engineered peptide

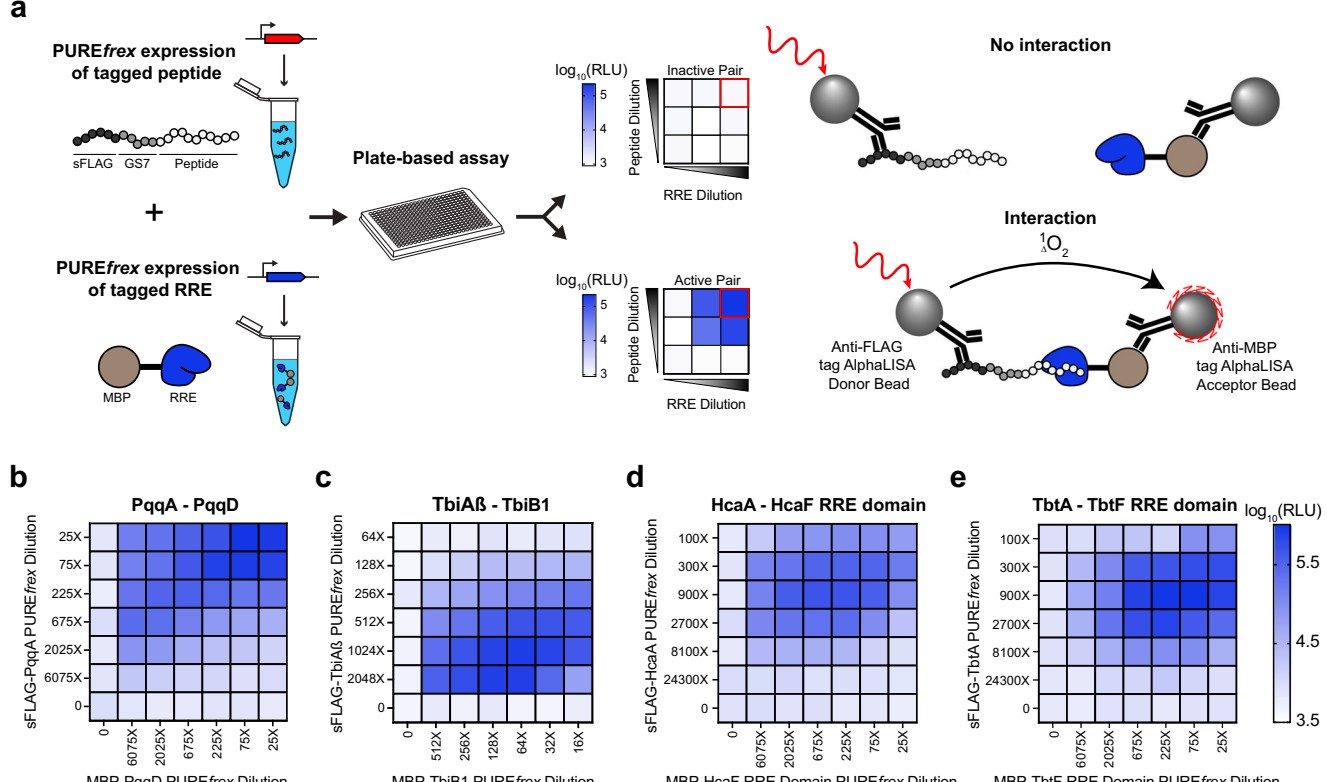

**Fig. 1 | A cell-free plate-based assay for detecting RRE-peptide interactions.**
**a** Schematic of the cell-free workflow. sFLAG-tagged peptides and MBP-tagged RREs are expressed in individual PURE*frex* reactions, mixed in a 384 well plate, and incubated to enable binding interactions. Addition of anti-FLAG AlphaLISA donor beads and anti-MBP AlphaLISA acceptor beads enables detection of interactions between the RRE and peptide of interest. PURE*frex* reactions of precursor peptide and RRE for (**b**) pyrroloquinoline quinone (PQQ), (**c**) a putative lasso peptide from

*Thermobacculum terrenum* ATCC BAA−798, (**d**) a heterocycloanthracin from *Bacillus* sp. Al Hakam, and (**e**) thiomuracin, a thiopeptide from *Thermobispora bispora* were cross-titrated across different dilutions (with 0 indicating no PURE*frex* reaction added) and assessed for binding interactions via AlphaLISA. Data are representative of three (*n* = 3) biological replicates. RLU relative luminescence units. Source data are provided in the Source Data 1 file.

variant (peptide variant 3) and TbtF. Next, we created peptide variants 4–10 by adding individually, or in combination, residues L(-34), P(-28), and M(-22), which in our initial screen also appeared to slightly reduce binding affinity to TbtF when mutated to an alanine. Adding both P(-28) and M(-22) (peptide variant 9) to peptide variant 3 enabled weak binding by TbtF, with ~25% AlphaLISA signal of the wild-type TbtA leader sequence. Further addition of residues resulted in a synthetic peptide (peptide variant 12) that is 40% identical to the leader sequence of TbtA (L(-34), N(-33), L(-32), D(-30), L(-29), P(-28), M(-27), D(-26), F(-24), E(-23), and M(-22)) and exhibits binding to TbtF (AlphaLISA signal) that is approximately equal to that observed with the wild-type TbtA leader sequence peptide. Interestingly, adding residues D(-20) and S(-31) (peptide variant 14) increased the signal further to ~2-fold higher than that observed with the wild-type TbtA leader sequence. These results highlight our assay's ability to rapidly identify specific residues involved in RRE-peptide binding interactions and design peptide sequences with the minimum number of residues required for RRE engagement.

### Screening computationally identified RRE-peptide pairs

We next wanted to show how our workflow could characterize RRE binding for BGCs computationally predicted via AntiSMASH[69]. Successful heterologous expression of computationally predicted RiPP products in vivo can be a challenge due to the inability to precisely control expression timing and yield, as well as the absence of necessary cofactors[70]. We chose to study lasso peptides due to their unique lariat structure which imparts the molecule with a wide range of beneficial characteristics, such as heat and protease stability[71,72], and because

they have been successfully expressed before in cell-free systems[73]. Additionally, lasso peptides have displayed a variety of bioactivities, including antimicrobial activity[74-79]. Biosynthetically, lasso peptide BGCs typically encode (i) a precursor peptide, (ii) an RRE and (iii) a protease, or a fusion protein encoding both the RRE and protease, as well as (iv) a cyclase[71]. In all reported lasso peptide BGCs, RREs are important for guiding the protease to the precursor peptide substrate and in some cases are also required for cyclization by the cyclase[26,80-84].

To begin, we used AntiSMASH[69] to identify a total of 2,574 lasso peptide BGCs from a collection of 39,311 diverse genomes (Fig. 3a; Supplementary Table 2). Of these, 1,882 BGCs were predicted to contain all essential lasso peptide biosynthetic enzymes (Supplementary Table 3). We compared the identified BGCs to known lasso peptides by constructing a sequence similarity network of the predicted core peptide sequences and annotating known sequences within the resulting network. Sequences that matched computationally predicted but not experimentally verified sequences reported in the literature were maintained in the dataset while those that had been characterized experimentally at the time of our analysis were removed; in doing so, we reasoned that our workflow could help validate predictions generated by others in the field. From the remaining predicted BGCs, 47 were selected for study from 32 unique genera based on their potential for antibiotic activity (Supplementary Data 1).

Of the 47 predicted lasso peptide BGCs, 5 were predicted to contain more than one precursor peptide and/or RRE, bringing the total number of predictions to 57 unique precursor peptides and 52 unique RREs. We applied our cell-free workflow to screen all 57 predicted precursor peptides with their associated predicted RREs

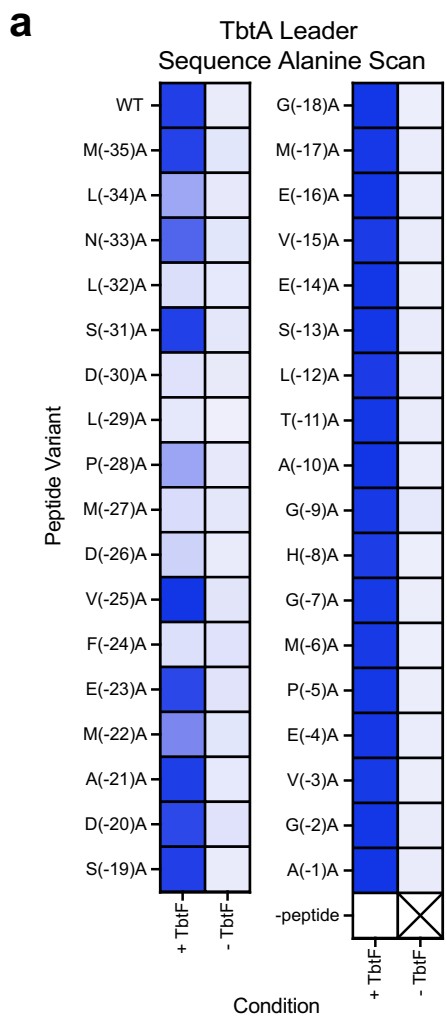

**a**

TbtA Leader
Sequence Alanine Scan

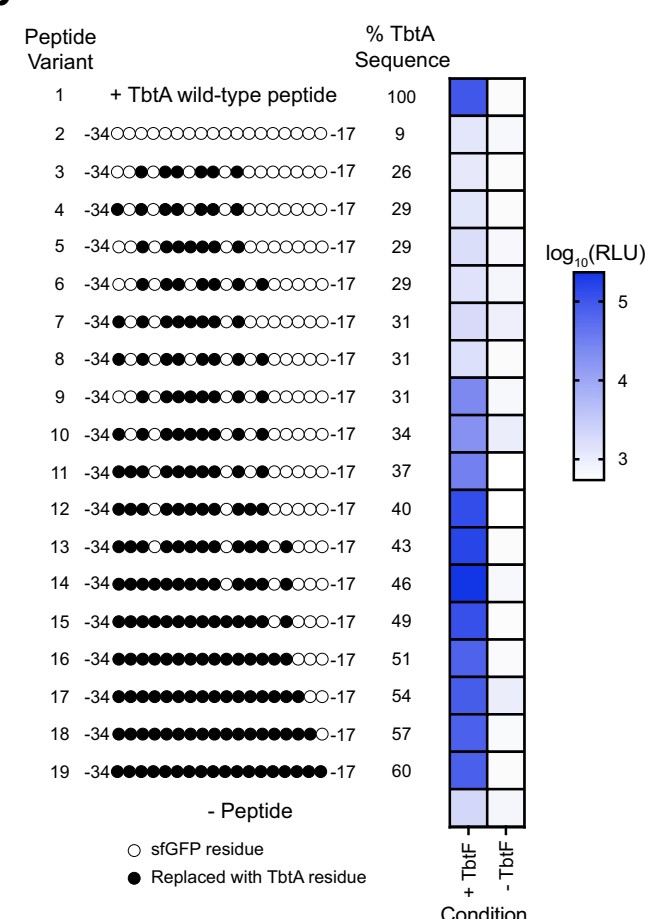

**b**

Fig. 2 | Cell-free workflow identifies peptide residues important for binding by TbtF. a An alanine scan library of the leader sequence of TbtA was expressed in individual PURE*frex* reactions and assessed for binding interactions in the presence of MBP-TbtF RRE domain using AlphaLISA. b A synthetic peptide library was constructed using the first 40 amino acids of sfGFP. Variants of the sfGFP were then constructed by replacing residues in the peptide identified in the alanine scan as important for binding by TbtF with the corresponding residue in the wild-type TbtA leader sequence. Each peptide variant was expressed in an individual PURE*frex* reaction and then assessed for binding interactions in the presence and absence of

TbtF using AlphaLISA. Peptide variant 2 contains 9% identity to TbtA wild-type peptide due to sharing residues G(-2), G(-7) and G(−9). For simplicity, only amino acids between the −34 and −17 position are depicted, however each peptide was composed of 40 amino acids reflecting the length of the TbtA leader sequence with an additional 5 amino acid linker. Sequences for each of the peptide variants assayed in panel b are provided in Supplementary Table 1. All data are presented as the mean of technical replicates (*n* = 3). RLU relative luminescence units. Source data are provided in the Source Data 1 file.

(Fig. 3b, c; Supplementary Fig. 6). To account for potential differences in expression levels in the PURE*frex* reactions as well as the fact that RREs reported in literature have a range of binding affinities, we tested each peptide-RRE pair at multiple concentrations. In instances where multiple RREs or precursor peptides were predicted in the same BGC, we screened all pairwise combinations. In total, we screened 72 different RRE-peptide pairs, 42 RRE-peptide pairs from clusters with a single predicted RRE and peptide pair (Fig. 3b), and 30 different combinations of RRE and peptides from clusters with multiple predicted genes for each (Fig. 3c; Supplementary Fig. 6).

Our initial screen yielded clear binding patterns for 27 of the 42 individual RRE-peptide pairs and 24 of the 30 RRE-peptide combinations from larger clusters (Fig. 3b, c; Supplementary Fig. 6). A subsequent validation experiment assaying all RRE-peptide pairs in biological triplicate at the dilution condition that yielded the highest AlphaLISA signal confirmed the results of our screen, with RRE-peptide pairs that produced higher AlphaLISA signal in our initial screen generally producing higher AlphaLISA signal in the validation experiment

(Supplementary Fig. 7). Notably, in addition to identifying functional RREs and peptide pairs, our methodology enables the rapid testing of more complex BGCs. For example, 44B1-1 can bind to both precursor peptides identified in the BGC while 44B1-2 can only bind to the second precursor peptide (Fig. 3c). Similar behavior emerged in BGC 46 in which predicted RREs bound to both, only one, or neither of the precursor peptides (Fig. 3c).

Using the results from our large-scale RRE screen, we prioritized BGCs identified as "hits" for complete biosynthesis of a mature lasso peptide in vitro. To do so, we expressed precursor peptides in PURE-*frex* reactions and purified each related tailoring enzyme heterologously expressed in *Escherichia coli*. Small scale (10 µL) reactions were assembled by combining precursor peptides and purified tailoring enzymes and analyzed via matrix-assisted laser desorption/ionization time-of-flight (MALDI-TOF) MS after overnight incubation at 37 °C. By testing 24 clusters, we successfully produced one peptide with the topology of a lasso peptide from BGC 24 (Supplementary Figs. 10 and 11). Subsequent characterization experiments confirmed

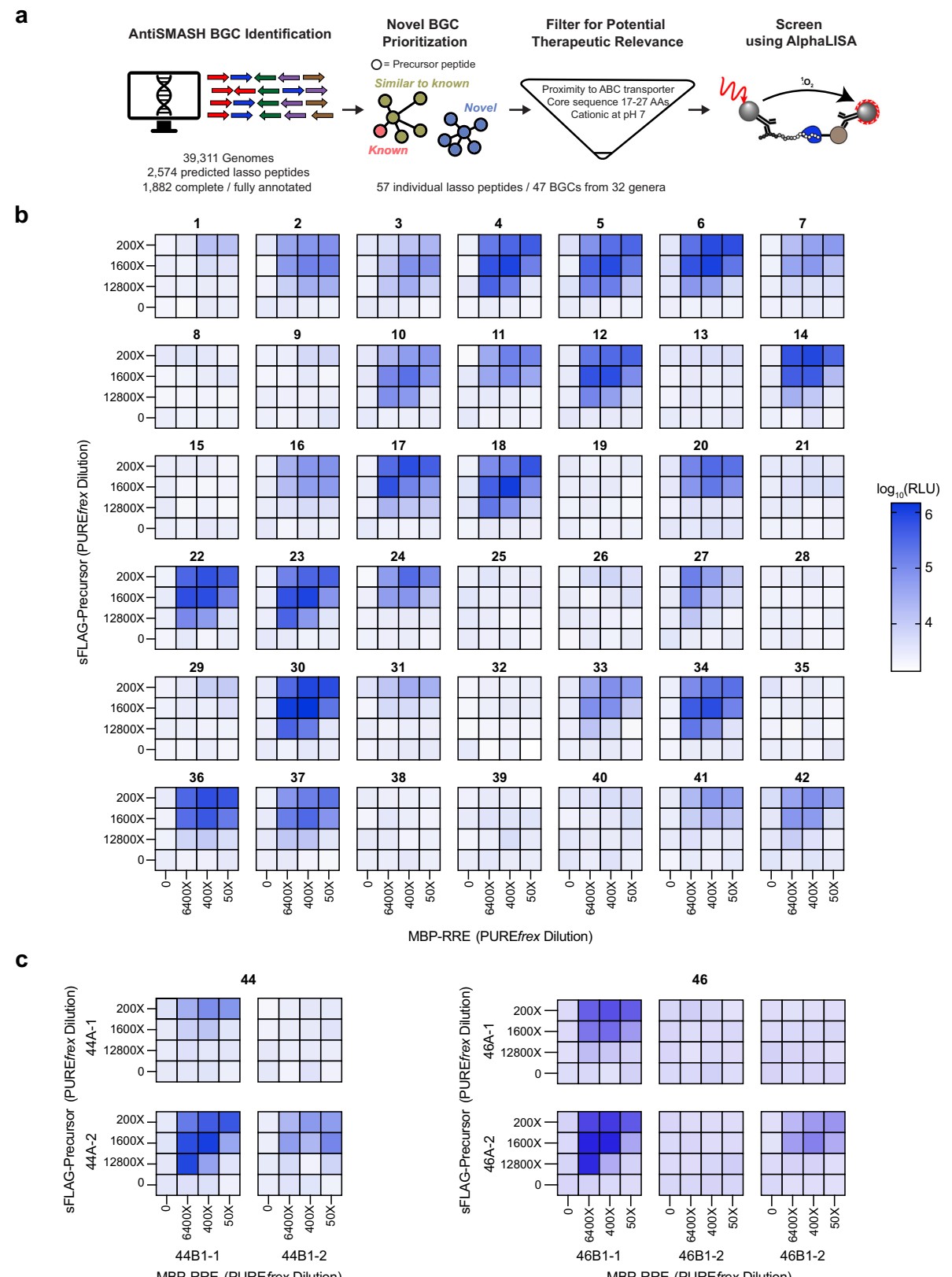

**Fig. 3 | Computationally guided screen of lasso peptide RREs. a** Overall prediction and screening workflow for lasso peptide BGCs. **b** For predicted BGCs with a single RRE and precursor peptide, individual PURE*frex* reactions expressing RREs and the respective peptide were cross-titrated and assessed for binding activity via AlphaLISA. **c** For predicted biosynthetic gene clusters with multiple RREs and precursor peptides, all possible combinations of RRE and precursor peptide were assessed for binding activity via AlphaLISA (select combinations shown, see Supplementary Fig. 6 for additional pairwise combinations). All data shown are single replicate, with confirmation reactions performed in biological triplicate provided in Supplementary Fig. 7 to show reproducibility. RLU relative luminescence units. Source data are provided in the Source Data 1 file.

that the production of this lasso peptide, Las24, is time dependent (Supplementary Fig. 12), that all proteins in the predicted BGC are necessary for maturation (Supplementary Fig. 13), that the sequence of the molecule matches the expected structure (Supplementary Fig. 14), that the molecule is resistant to carboxypeptidase (a common confirmation of threaded topology) (Supplementary Fig. 15), and that there is limited to no interaction of Las24 biosynthetic components with those from other lasso peptide BGCs (Supplementary Fig. 16). During our work, King et al. reported the heterologous production of this lasso peptide (termed Las-1010) in *E. coli* from the same biosynthetic cluster[85]. Las-1010, which was previously bioinformatically identified[86] but not experimentally characterized, was found by King et al. to exhibit weak antibacterial activity against some bacterial strains[85]. Taken together, we showed that the integration of CFE and AlphaLISA allows for rapid prototyping of RiPP BGCs by detecting RRE-peptide interactions, mapping RRE-peptide binding landscapes, and screening computationally identified RRE-peptide pairs.

## A cell-free AlphaLISA-based workflow for prototyping in vitro glycosylation reactions

We next showed the generalizability of our cell-free AlphaLISA workflow by exploring an important PTM in protein biologics, namely glycosylation. To show this, we chose to study the activity of oligosaccharyltransferases (OSTs) using bacterial glycans relevant to conjugate vaccine production. Conjugate vaccines, composed of a pathogen-specific polysaccharide antigen (e.g., O-antigen polysaccharide or capsular polysaccharide (CPS)) linked to an immunogenic carrier protein, are a promising strategy to protect against bacterial infections[87]. Both the glycan and carrier protein play an important role in developing long-lasting immunity[88].

A current challenge with manufacturing conjugate vaccines is the reliance on a multi-step process in which the glycan is isolated from the targeted pathogenic bacteria and chemically conjugated to a separately produced carrier protein[89]. A related limitation is the lack of control over the site of glycan attachment using traditional chemical conjugation techniques[90]. To address these challenges, recent work in the fields of glycobiology and synthetic biology has developed both cell[91–96] and cell-free[97–100] based methods for producing conjugate vaccines using OSTs to site-specifically transfer glycans onto a carrier protein, called protein glycan coupling technology[101,102].

We modified our workflow used with RiPPs to characterize glycosylation of carrier proteins using OSTs. We first express our carrier protein in a standard CFE reaction and our OST in a CFE reaction supplemented with nanodiscs, which act as membrane mimics into which membrane-bound proteins can express solubly[103]. We next mix CFE-expressed OST, CFE-expressed carrier protein, and a crude membrane fraction enriched with bacterial glycan to assemble an in vitro glycosylation (IVG) reaction (Supplementary Fig. 18a). The crude membrane fraction is produced from *E. coli* cells expressing a single biosynthetic pathway encoding a single pathogen-specific O-antigen or capsular polysaccharide. Glycosylation with that polysaccharide can then be detected with AlphaLISA (Fig. 4a).

As a model system to demonstrate this workflow, we selected the capsular polysaccharide from *Streptococcus pneumoniae* serotype 4 (CPS4) as the pathogen glycan. The CPS4 glycan is composed of the repeating tetrasaccharide unit PyrGal-ManNAc-FucNAc-GalNAc (PyrGal: pyruvate attached to galactose; ManNAc: *N*-acetylmannosamine; FucNAc: *N*-acetylfucosamine; GalNAc: *N*-acetylgalactosamine) and is important for conjugate vaccine protection against pneumococcal infection[104]. Other groups have previously shown that the CPS4 glycan can be synthesized in strains of *E. coli* via recombinant expression of the CPS4 biosynthetic pathway and attached to proteins using the OST PglB from *Campylobacter jejuni* (*Cj*PglB)[104–106]. We began by overexpressing the CPS4 glycan in *E. coli* cells, harvesting and lysing the cells, and concentrating the membrane vesicles containing CPS4 via ultracentrifugation to produce CPS4-enriched crude membrane fractions (CMFs). Following verification of the presence of the CPS4 glycan in our CMF with an anti-CPS4 dot blot (Supplementary Fig. 19), we then showed that we can perform glycosylation in IVG reactions using the CPS4 glycan. Using a 6xHis tag on the carrier protein, we performed Western blot analysis to confirm transfer of the targeted bacterial glycan onto the protein, with the banding pattern above the aglycosylated protein corresponding to transfer of different chain lengths of the bacterial glycan (Supplementary Fig. 18b). IVG reactions using CMF prepared from cells without CPS4 overexpression confirmed that only CPS4 is being transferred onto the carrier protein in our system, while an anti-CPS4 Western blot confirmed the identity of the CPS4 glycan on our glycoconjugates (Supplementary Fig. 20).

We then asked whether we could adopt our cell-free AlphaLISA workflow to detect glycosylation. We hypothesized that we would be able to distinguish between glycosylated and aglycosylated proteins by incorporating anti-glycan serum antibodies into the AlphaLISA reaction and using Protein A AlphaLISA donor beads and anti-6xHis AlphaLISA acceptor beads. Indeed, when we prepared IVG reactions using an acceptor protein containing a sequon (a short sequence of amino acids) that can (DQNAT) or cannot (AQNAT) be glycosylated by *Cj*PglB and analyzed the reactions using AlphaLISA, we observed a distinct binding pattern only when we use the protein containing DQNAT, confirming our ability to discriminate between glycosylated and aglycosylated samples (Fig. 4b, c).

## Cell-free workflow enables engineering of *Cj*PglB for increased transfer efficiency of CPS from *S. pneumoniae* serotype 4

We next sought to determine if we could use our workflow to identify OST variants that have improved glycan transfer efficiency. While *Cj*PglB has demonstrated glycan substrate promiscuity, the efficiency with which it can glycosylate acceptor proteins with different glycans varies widely[91–93,95]. To address this challenge, recent work has demonstrated that mutating PglB can lead to improvements in glycosylation efficiency[94,107]. When we tested two previously identified *Cj*PglB mutants, we observed improvements in glycosylation efficiency with the CPS4 glycan (Supplementary Fig. 18b). To improve glycosylation efficiency further, we designed a mutant library of *Cj*PglB to test via AlphaLISA.

We identified 15 *Cj*PglB residues for site saturation mutagenesis: 9 residues (Y77, S80, S196, N311, Y462, G476, G477, H479, and K522) based on their predicted location within 4 angstroms of where the innermost sugar of the native *Cj*PglB glycan sits within the enzyme's active site, 3 residues (Q287, L288, and K289) within external loop 5 (EL5) that have previously been shown to be highly mutatable, and 3 additional residues (D475, K478, and L480) located in a flexible loop located directly above the nitrogen atom of the amide group on the acceptor protein where the glycan is covalently linked[107,108] (Fig. 5a, b). Our library of *Cj*PglB mutants contains each of these 15 residues individually mutated to all 19 other amino acids, resulting in a set of 285 unique single mutant *Cj*PglB constructs along with the wild-type sequence.

Using our cell-free workflow, we rapidly expressed the complete mutant library and assayed for activity (Fig. 5c). Ten *Cj*PglB mutants (S80V, S80T, Q287K, N311I, N311V, N311M, L480A, L480W, and L480R) produced higher AlphaLISA signal than the WT *Cj*PglB construct. Most sites were inflexible to mutation and produced no hits, whereas the sites S80, N311, and L480 produced multiple high-signal mutants. Control reactions that contained all reaction components except the CPS4 antiserum produced AlphaLISA signal equivalent to background (Supplementary Fig. 21a), and duplicate measurements for each *Cj*PglB mutant were consistent (Supplementary Fig. 21b).

To validate the results of our screen, we performed Western blot analysis of IVG reactions glycosylating the clinically relevant carrier protein *Haemophilus influenzae* protein D (PD) with CPS4 using each of

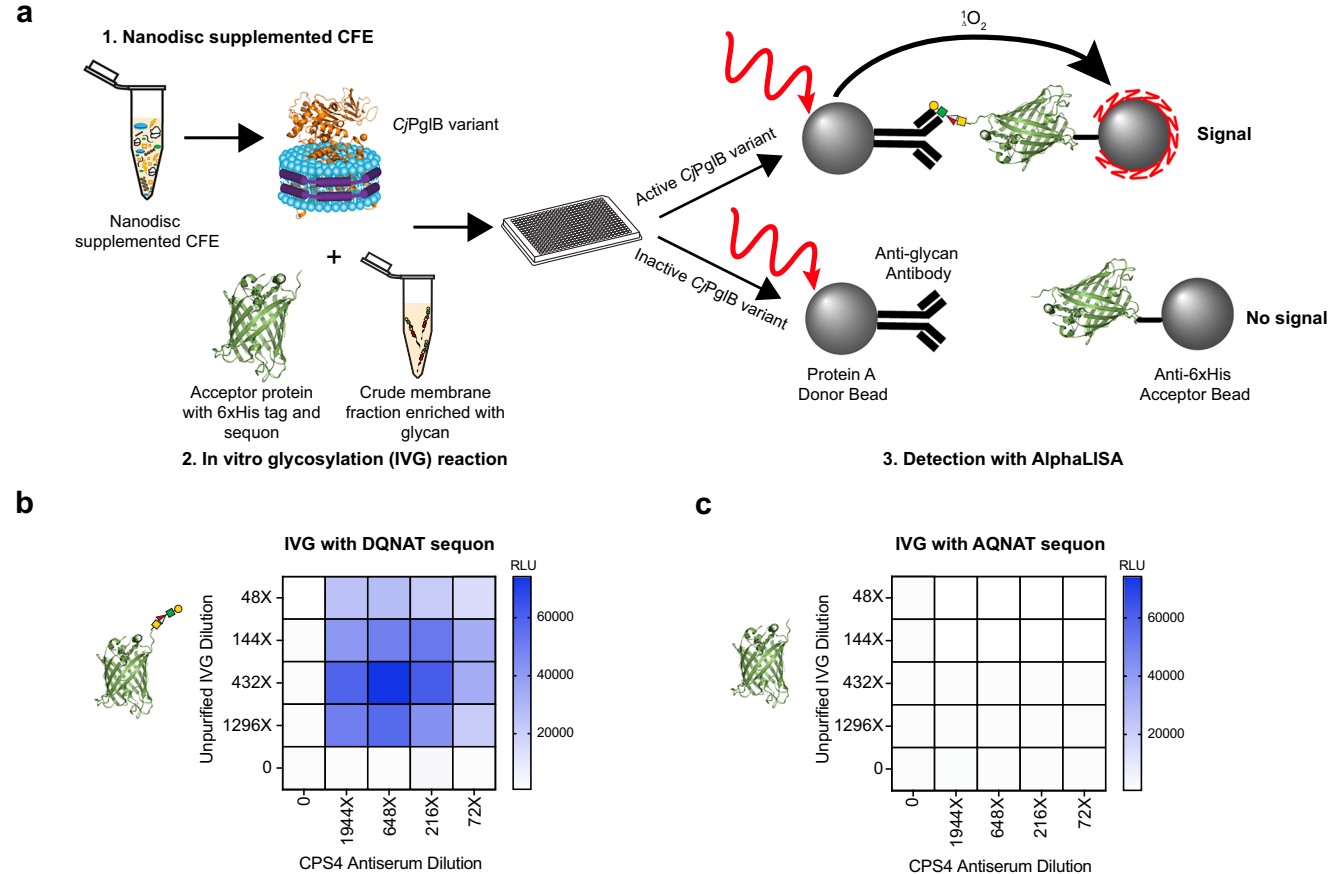

**Fig. 4 | CFE and AlphaLISA can be combined to prototype in vitro glycosylation reactions. a** Schematic of the cell-free workflow. Nanodisc supplemented CFE reactions were first used to express *Cj*PglB variants and then mixed with an acceptor protein containing a 6xHis tag and sequon and crude membrane fraction enriched with a bacterial glycan of interest. Samples were then analyzed using Protein A AlphaLISA donor beads and anti-6xHis AlphaLISA acceptor beads. **b** An IVG reaction using *Cj*PglB, crude membrane fraction enriched with CPS from *S. pneumoniae* serotype 4, and sfGFP with a 6xHis tag and either (**b**) a DQNAT sequon or (**c**) an AQNAT sequon was serially diluted, mixed with varying concentrations of *S. pneumoniae* CPS4 antiserum, and analyzed via AlphaLISA. All data are representative of three independent experiments (*n* = 3). RLU relative luminescence units. Source data are provided in the Source Data 1 file.

the seven highest-signal mutants (S80V, S80T, Q287K, N311I, N311V, N311M, and L480R) and compared the transfer efficiency to WT *Cj*PglB (Supplementary Fig. 22). Each mutant produced a higher transfer efficiency than the WT enzyme, with the PglB^Q287K variant raising the transfer efficiency by 38% (-1.7x) to an efficiency of 91%. An anti-CPS4 Western blot confirmed transfer of the CPS4 glycan using two top *Cj*PglB hits (Supplementary Fig. 20).

### Cell-free workflow enables rapid identification of sites accessible for glycosylation in in vitro glycosylation reactions

With an efficient OST in hand, we next asked at which locations throughout a model vaccine carrier protein we could attach the bacterial polysaccharide. Conventional technologies to produce conjugate vaccines use chemical methods to randomly conjugate glycans to a carrier protein[89], which can be inefficient due to accessibility of the glycan attachment site and reduce vaccine immunogenicity[109,110]. In comparison, enzymatic production of conjugate vaccines using OSTs enables site-specific glycosylation of a carrier protein precisely at the synthetically inserted sequon, which could be exploited to achieve high levels of protein expression as well as efficient and efficacious glycosylation[111–113]. However, thus far, cell-free approaches to produce conjugate vaccines have typically relied on placing the sequon at the C-terminus of carrier proteins[97–99], with limited exceptions[97].

We sought to discover which sites within a clinically relevant carrier protein can be efficiently, enzymatically glycosylated in vitro.

We therefore used our workflow to screen a comprehensive library of carrier protein constructs containing a sequon placed between every pair of amino acids throughout the carrier protein PD (Fig. 6a)[114,115].

Our library contained 328 unique PD sequences in which the glycosylation sequon "DQNAT," surrounded by short linkers, was placed between every two amino acids in the carrier protein, beginning with an N-terminal sequon placement and ending with a C-terminal sequon placement. We then applied our cell-free workflow to this library by synthesizing each construct in a CFE reaction, combining each synthesized carrier with CPS4 glycan and *Cj*PglB^Q287K–the high efficiency mutant identified in the OST mutagenesis screen–and assessing glycosylation of each carrier protein construct in parallel in 1-µL AlphaLISA reactions using conditions optimized to detect low glycosylation levels (Fig. 6b; Supplementary Fig. 23).

Our screen identified three sections of PD that were amenable to glycosylation in IVG reactions: 32 sites at the N-terminal end of the carrier sequence, a stretch of -20 internal sites, and 40 sites at the C-terminal end of the carrier sequence (Fig. 6b). Mapping AlphaLISA signal to a crystal structure of PD reveals one 3-dimensional section of the carrier protein that is able to be glycosylated (Fig. 6c)[116]. In total, 94 sequon positions showed statistically significant signal above a negative control containing the sequon "DQLAT" (Fig. 6b; Supplementary Figs. 24a and 25), with top performing variants producing signal >100x above background. Triplicate measurements for each sequon variant were consistent with each other, and all negative

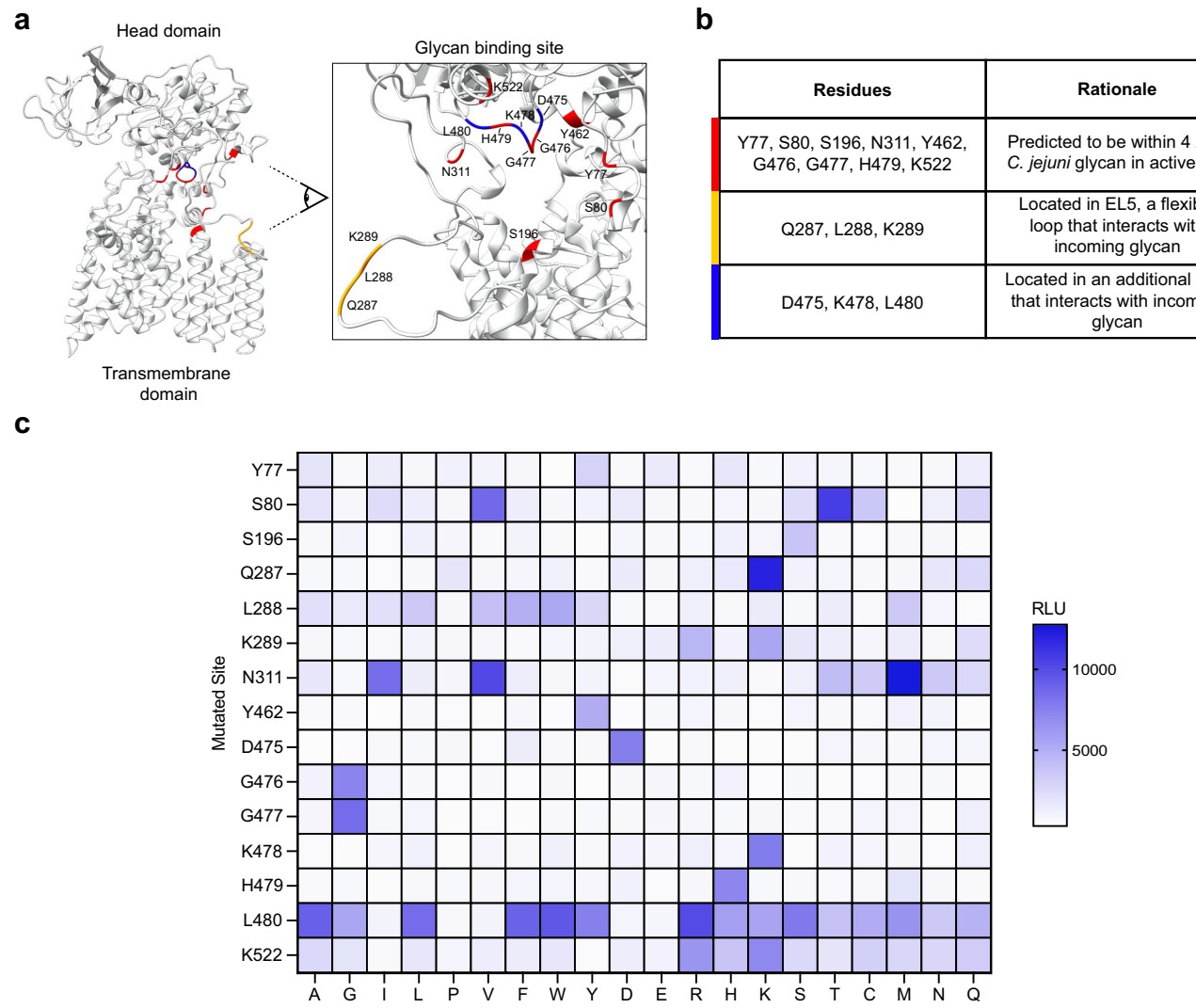

**Fig. 5 | Cell-free workflow identifies high-efficiency *Cj*PglB mutants. a** Homology model of *Cj*PglB[94]. Sites chosen for site saturation mutagenesis are highlighted in colors denoted in (**b**). **b** Rationale of sites chosen for mutagenesis. **c** AlphaLISA results for IVG reactions containing crude membrane fraction enriched with CPS from *S. pneumoniae* serotype 4, sfGFP with a 6xHis tag and DQNAT sequon, and a unique *Cj*PglB mutant. Data are a mean of AlphaLISA signals produced by duplicate IVG reactions (*n* = 2). RLU relative luminescence units. Source data are provided in the Source Data 1 file.

control variants had signal equivalent to background (Supplementary Figs. 24b and 25). A selection of sequon variants with high and low AlphaLISA signal were validated via Western blot, confirming the accuracy of our workflow for comparing glycosylation efficiency of different carrier protein constructs (Supplementary Fig. 26).

## Discussion

In this work, we established an integrated workflow for expressing and characterizing proteins involved in PTM installation. This workflow uniquely combines methods for cell-free DNA assembly and amplification, cell-free gene expression, and binding characterization via AlphaLISA. We show that the platform is generalizable, fast (steps are carried out in hours), and readily scalable to 384- or 1536-well plates without the need for time intensive protein purification or cell-based cloning techniques. Moreover, the platform is designed with automation in mind, with each step consisting of simple liquid handling and temperature incubation steps. We showed the utility of the platform for characterizing the activity of RREs involved in RiPP biosynthesis as well as towards engineering systems for efficient conjugate vaccine

production, including protein engineering to increase the efficiency of a glycan-installing enzyme.

Through our work characterizing the binding activity of TbtF to TbtA, we found that our methodology can within hours of obtaining DNA samples recapitulate findings obtained using traditional approaches that take days to weeks to perform. Looking forward, we can use our workflow for more advanced PTM engineering strategies. For example, recent efforts have created novel RiPP products by engineering peptide substrates to contain leader sequences recognized by tailoring enzymes from multiple classes of RiPPs[117]. In doing so, a peptide substrate was modified with RRE-dependent tailoring enzymes from two different BGCs. Creating more complex systems with even greater numbers of RRE-dependent modifications will require an understanding of appropriate design rules for enabling recognition of the precursor peptide by the desired tailoring enzymes. Using the information gained by mutational scanning, we were able to systematically produce a synthetic peptide with only 40% identity to the wild-type peptide that exhibits AlphaLISA binding signal on par with the wild-type peptide. Understanding the minimal set of amino acid

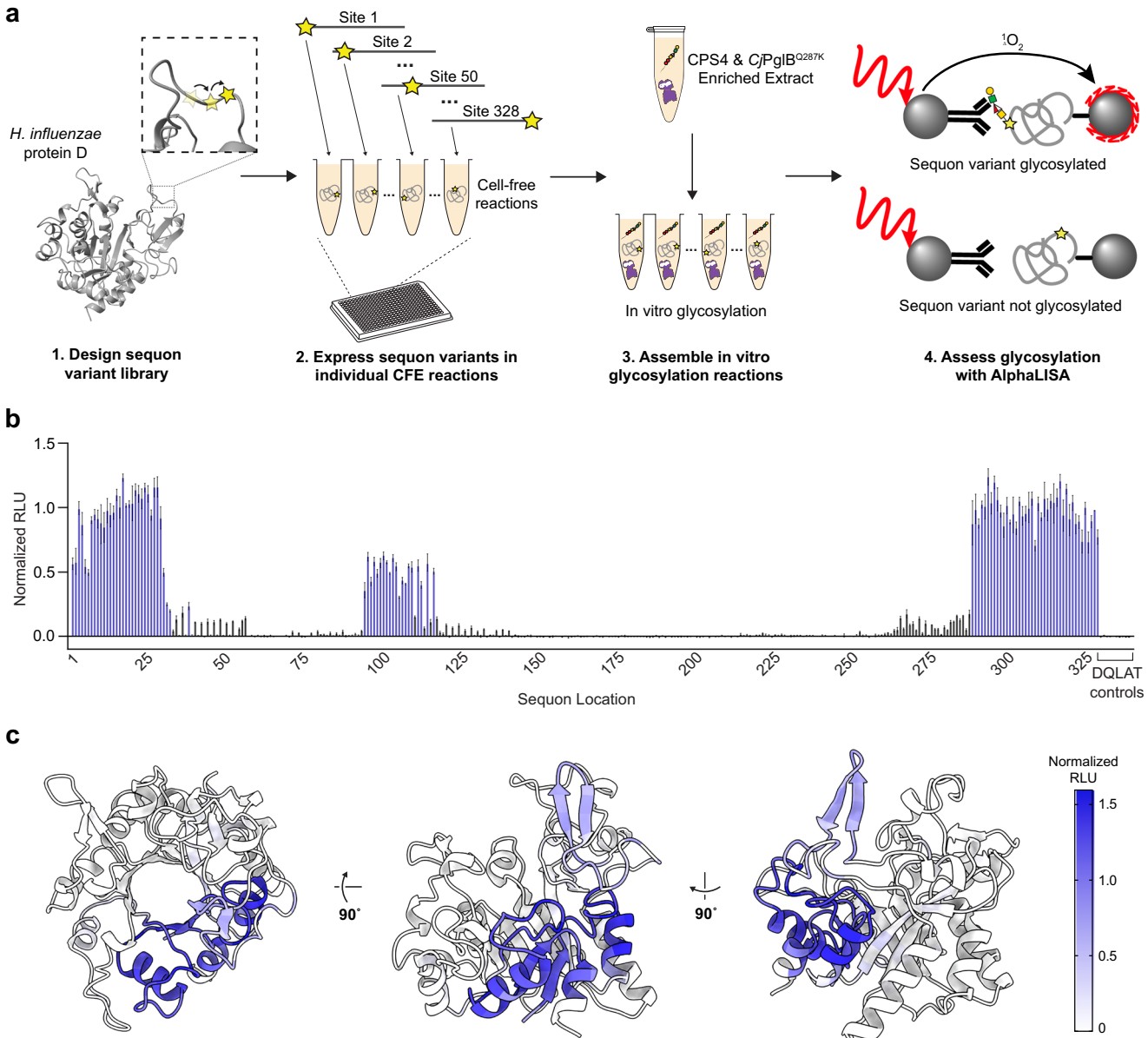

**Fig. 6 | Sequon scanning of *H. influenzae* protein D. a** Schematic of the cell-free workflow. A library of PD constructs was designed with a glycosylation sequon inserted between every two amino acids. Each sequon variant was expressed in an individual CFE reaction. Following expression, sequon variants were combined with extract enriched with *S. pneumoniae* CPS4 glycan and *Cj*PglB[Q287K] to form IVG reactions. IVG products were assessed for glycosylation with AlphaLISA. **b** AlphaLISA results for sequon scanning from N-terminus (site 1) through C-terminus (site 328). Data are presented as the mean of $n = 3$ biological replicates. "DQLAT" sequons, due to the lack of an asparagine residue, are unable to be *N*-glycosylated by *Cj*PglB. Blue bars denote sequon positions resulting in significantly higher AlphaLISA signal than the negative control, as determined using one-way ANOVA with Bonferroni correction. Error bars show SEM. **c** AlphaLISA results from (**b**) mapped onto the crystal structure of PD. RLU relative luminescence units. Source data are provided in the Source Data 1 file.

residues required for recognition will be important for engineering increasingly complex molecules. Our workflow provides a method for understanding and prototyping these requirements.

Additionally, by coupling our workflow with computational prediction tools, we demonstrated how our platform can screen for natural product BGCs likely to function in an in vitro setting. While we were able to produce Las-1010[85] (Las24), many of the clusters we detected binding activity for in our screen did not produce mature lasso peptides. We hypothesize that this could be due to a number of reasons, including that the in vitro reaction environment may lack other important components natively present in in vivo systems, such as auxiliary genes as has been demonstrated for other natural products[118,119]. Advances in developing cell-free lysates from non-*E. coli*-based organisms, such as *Streptomyces*[120–123], could be incorporated

into future studies as a method for testing expression systems that contain native auxiliary factors. Despite this, by prioritizing BGCs with a demonstrated functional first step (RRE binding), our methods can be used as is to narrow down the number of proteins needed to be expressed and purified for attempts at in vitro reconstitution. Furthermore, we note that recent advances in deep learning models[124,125] and protein language models[126] have accelerated our ability to predict the substrate promiscuity of RiPP biosynthetic enzymes. Key to these computational tools and future Artificial Intelligence (AI) models is the ability to rapidly generate training datasets and validate any resulting predictions. Our workflow can be interfaced with these computational tools.

We also demonstrated how our workflow can be used to engineer both enzymes and substrates used in glycoprotein synthesis systems.

With our platform, we rapidly assessed 285 unique mutants of *Cj*PglB to enable efficient production of glycoproteins with the CPS from *S. pneumoniae* serotype 4, a major cause of pneumonia in disadvantaged communities[127]. Importantly, our screen uncovered beneficial mutations in both undiscovered sites and sites that had been identified in previous *Cj*PglB mutagenesis experiments[108] for other unrelated glycans, demonstrating the significance of a fast, high-throughput method to discover mutations unique for transferring each pathogen glycan of interest[94]. Because the identity of all tested mutants, including both low- and high-performing mutations, is known at the time of assay, we believe our workflow could be readily interfaced with machine-learning guided strategies[128–130] to more rapidly engineer oligosaccharyltransferases.

To highlight the future potential for making and optimizing conjugate vaccines, we used our workflow to rapidly assess the glycosylation of 328 unique variants of the carrier protein PD in vitro to discover high efficiency glycosylation sites throughout the carrier protein. Over a quarter of all sites produced AlphaLISA signal significantly higher than a negative control. Mapping sites that produced high AlphaLISA signal to a crystal structure of PD revealed one section of the protein that was highly glycosylated, suggesting that steric effects may play a role in determining the ability to glycosylate unique sequon positions[131] or that placing sequons in other locations of the protein may lead to low protein expression in IVG reactions.

One limitation of our work is that the results we obtain are semi-quantitative. With our current platform design, we can compare the relative binding affinity of different RRE-peptide pairs or glycosylation efficiency of specific OST mutants but are unable to provide exact quantitative measurements of these phenomena (e.g., $k_d$, % glycosylation, etc.). Thus, we suggest that our method can be integrated with more traditional assays by first using our workflow as a screening tool to down-select specific protein variants from larger libraries for follow-up experiments with smaller numbers of samples.

In sum, we developed a versatile, rapid, and robust cell-free platform for characterizing and engineering PTMs. We expect that this platform can be applied to other classes of PTMs and will accelerate the design and production of biologics with complex PTMs and improved therapeutic properties.

## Methods

### DNA design and preparation for RiPPs

For the initial screen of known RRE's, gene constructs were ordered from Twist Biosciences (synthesized into pJL1 backbone between NdeI and SalI restriction sites). Briefly, sequences were retrieved from literature or Uniprot and codon optimized using the IDT Codon Optimization Tool. For full length RRE constructs, a codon optimized sequence for a Twin-Strep tag and PAS11 linker were added to the N-terminus of the nucleotide sequence. MBP-fusion RRE constructs were constructed by replacing either the C-terminus (for proteins in which the RRE domain was predicted to occur in the N-terminus) or N-terminus (for proteins in which the RRE domain was predicted to occur in the C-terminus) portion of the sequence with codon optimized sequences for MBP and a GS7 linker. For precursor peptide sequences, sequences encoding either the full-length precursor or leader sequence were fused to an N-terminal sFLAG tag and GS7 linker.

For all peptide sequences used in AlphaLISA based assays, an N-terminal sFLAG tag and GS7 linker were incorporated into the design. For sequences utilized in the AlphaLISA alanine scan workflow, each alanine variant peptide was constructed by replacing the corresponding wild-type codon with "GCC". To construct synthetic sfGFP peptides, the first 40 amino acids of sfGFP (with a G23T mutation) was first codon optimized. Each variant was then constructed by replacing the appropriate wild-type codon with the codon corresponding to the desired residue change. All peptide sequences were ordered as eBlocks

with overhang to a linearized pJL1 backbone for use in Gibson Assembly reactions.

For all computationally predicted lasso peptide proteases and cyclases, the predicted gene sequences were codon optimized using the IDT Codon Optimization Tool. At the N-terminus of each sequence, maltose binding protein (MBP) and a short linker were incorporated to enable soluble expression and detection via AlphaLISA based assays. All genes were synthesized by Twist Biosciences either in pJL1 (for expression in PURE*frex*) or in a modified pET vector (for in vivo expression). The corresponding (untagged) precursor sequences were also synthesized by Twist Biosciences in pJL1 for use in assembling complete lasso peptide BGCs.

DNA templates for expression in PURE*frex* were prepared either in plasmid form using ZymoPURE II Plasmid Midiprep Kit (Zymo Research) or as linear expression templates (LETs). For LETs, eBlocks were inserted into pJL1 using Gibson Assembly with a linearized pJL1 backbone. Following Gibson Assembly, each reaction was then diluted 10x in nuclease free water. 1 μL of diluted Gibson Assembly reaction was then used in a 50 μL PCR reaction using Q5 Hot Start High-Fidelity DNA Polymerase (New England Biolabs).

All nucleotide sequences used in the RiPPs portion of this study are provided in the Supplementary Information or in the Supplementary Data 1 File.

### FluoroTect™ gel

PURE*frex* 2.1 (Gene Frontier) reactions were assembled according to manufacturer instructions using 1 μL of unpurified template LET and 0.5 μL of FluoroTect™ (Promega) per 10 μL reaction. Following incubation at 37 °C for 6 h, samples were centrifuged at 12,000 x *g* for 10 min at 4 °C. 3 μL of supernatant was then mixed with 1 μL of 40 μg/mL RNase A and incubated at 37 °C for 10 min. Following incubation, 1 μL of 1 M DTT, 2.5 μL of 4X Protein Sample Loading Buffer for Western Blots (Li-COR Biosciences), and 2.5 μL of water were added to each sample and the samples were then incubated at 70 °C for 10 min. Samples were then loaded on a NuPAGE 4–12% Bis-Tris Protein Gel and run for 40 min at 200 V in MES Running Buffer. For comparison, a lane was loaded with BenchMark fluorescent protein standard (Thermo Fisher Scientific). The resulting gel was then imaged using both the 600 and 700 fluorescent channel on a LICOR Odyssey Fc (Li-COR Biosciences).

### AlphaLISA reactions for RiPPs

PURE*frex* 2.1 (Gene Frontier) reactions were assembled according to manufacturer instructions. Briefly, 1 μL of the unpurified LET reaction—encoding for the precursor peptide or RRE—was added as a template per 10 μL PURE*frex* reaction. Reactions were then incubated at 37 °C for 5 h. After incubation, these samples were then diluted in a buffer consisting of 50 mM HEPES pH 7.4, 150 mM NaCl, 1 mg/mL BSA, and 0.015% v/v Triton X-100. Following dilution, an Echo 525 acoustic liquid handler was used to dispense 0.5 μL of diluted RRE, 0.5 μL of diluted peptide, and 0.5 μL of blank buffer from a 384-well polypropylene 2.0 Plus Source microplate (Labcyte) using the 384PP_Plus_GPSA fluid type into a ProxiPlate-384 Plus, White 384-shallow well destination microplate (Revvity). The plate was then sealed and equilibrated at room temperature for 1 h. Next, anti-FLAG Alpha Donor beads (Perkin Elmer) were used to immobilize the sFLAG tagged peptides and anti-Maltose-Binding (MBP) AlphaLISA acceptor beads were used to immobilize the MBP-tagged RREs. 0.5 μL of acceptor and donor beads diluted in buffer were added to each reaction to a final concentration of 0.08 mg/mL and 0.02 mg/mL donor and acceptor beads, respectively. Reactions were then equilibrated an additional hour at room temperature in the dark. For analysis, reactions were incubated for 10 min in a Tecan Infinite M1000 Pro (using Tecan i-control v. 3.9.1.0) plate reader at room temperature and then chemiluminescence signal was read using the AlphaLISA filter with an excitation time of 100 ms, an integration

time of 300 ms, and a settle time of 20 ms. Results were visualized using Prism version 9.5.1 (GraphPad).

## Computational prediction of lasso peptide BGCs

A diverse collection of 39,311 publicly available genomes (2020 April) spanning soil bacteria, metagenomes and extremophiles were analyzed using AntiSMASH 5.1.2 identifying 315,876 biosynthetic gene clusters (Supplementary Table 2). A total of 2,574 lasso peptide clusters were identified, and from this set we then performed an additional filtering step to identify 1,882 BGCs which contained a complete collection of essential biosynthetic enzymes (Supplementary Table 3). Specifically, we note that predictions using AntiSMASH rely on identifying clusters in which the predicted components include homology to PF13471 and a proximal asparagine synthetase, micJ25, or mcJC. Therefore, there is a possibility that clusters identified by AntiSMASH are missing essential enzymes (which we did observe) and we did not include these incomplete clusters in our follow-up analysis. To further prioritize these BGCs, a sequence similarity network[132,133] was used to group identified precursor peptides with a collection of known lasso peptide sequences. Peptide sequences that did not group with known sequences were considered novel and were nominated for further investigation. Subsequent filtering of the remaining novel BGCs included selecting BGCs based on a core peptide length of 17–27 amino acids and whether the mature lasso peptide is predicted to carry a positive charge at a neutral pH. Calculation of the predicted isoelectric point of the predicted core peptides used Thermo Fisher Scientific's peptide analysis tool (https://www.thermofisher.com/us/en/home/life-science/protein-biology/peptides-proteins/custom-peptide-synthesis-services/peptide-analyzing-tool.html). This narrowed the selection to 202 BGCs, of which 47 were chosen. A total of 210 genes were synthesized by Twist Bioscience. All amino acid sequences and metadata for the 47 selected BGCs are provided in the Supplementary Data 1 File.

## In vivo expression and purification of lasso peptide tailoring enzymes

For computationally predicted MBP-RREs and MBP-proteases, constructs of the target protein in pET.BCS.RBSU.NS backbone were transformed into BL21 Star (DE3) cells, plated on LB agar plates containing 100 µg/mL carbenicillin, and incubated at 37 °C. Single colonies were cultured in 50 mL of LB containing 100 µg/mL carbenicillin at 37 °C and 250 RPM. After overnight incubation, 20 mL of the overnight culture were used to inoculate 1 L of LB supplemented with 2 g/L glucose and 100 µg/mL carbenicillin. Cells were grown at 37 °C and 250 RPM and induced for protein production at $OD_{600}$ 0.6-0.8 with 500 µL of 1 M IPTG. Four hours post induction, cells were harvested via centrifugation at 5000 x $g$ for 10 min at 4 °C and flash frozen in liquid nitrogen.

After thawing on ice, cell pellets were resuspended in lysis buffer composed of 50 mM Tris-HCl pH 7.4, 500 mM NaCl, 2.5 % (v/v) glycerol, and 0.1% Triton X-100. For cell pellets used to overexpress RREs and cyclases, the lysis buffer also contained 6 mM PMSF, 100 µM Leupeptin, and 100 µM E64. Cell suspensions were then supplemented with 1 mg/mL lysozyme and lysed via sonication using a Qsonica sonicator at 50% amplitude for 2 min with 10 s on 10 s off cycles. Following sonication, insoluble debris were removed via centrifugation at 14,000 x $g$ for 30 min at 4 °C. Per 1 L of cell culture, 5 mL of amylose resin was equilibrated with 5 to 10 column volumes of wash buffer (50 mM Tris HCl, 500 mM NaCl, 2.5 % (v/v) glycerol, pH 7.4) in a 50 mL conical tube and mixed via inversion. Resin was separated from wash buffer by spinning at 2,000 x $g$ for 2 min at 4 °C and the supernatant was then poured off. Equilibration was repeated for a total of 4 times with fresh equilibration buffer. Following the last equilibration, the cleared cell lysis supernatant was added to the resin and incubated for 2 h at 4 °C with constant agitation on a shake table. Following

incubation on the resin, the resin was washed once with 5 column volumes of lysis buffer followed by 5 column volumes of wash buffer four times. For the last wash, the resuspended resin was loaded in a 25 mL gravity flow column and drained via gravity flow. For elution, 15 mL of elution buffer (50 mM Tris HCl, 300 mM NaCl, 10 mM maltose, 2.5% (v/v) glycerol, pH 7.4) was added to the gravity flow column and collected. Samples were then buffer exchanged into storage buffer (50 mM HEPES, 300 mM NaCl, 0.5 mM TCEP, 2.5% (v/v) glycerol, pH 7.5) using amicon spin filters (50 kDa MWCO) by spinning at 4,500 x $g$ for 10–15 min. Samples were then aliquoted, flash frozen, and stored at -80 °C until use. Total protein concentration of each purified sample was determined using a Bradford assay (Biorad). Percent purity of each sample was determined by running diluted aliquots of each purified protein on a 4–12% Bis-Tris gel and staining with Optiblot Blue (Abcam). After destaining, each gel was imaged using the 700 fluorescent channel on a LICOR Odyssey Fc (Li-COR Biosciences, USA) and percent purity was determined via densitometry using Licor Image Studio Lite (v. 5.2.5). Final concentrations of each protein were then calculated by multiplying the total protein content by the percent purity.

Computationally identified cyclases were expressed and purified according to the process outlined above for computationally identified RREs except for transforming into BL21 Star (DE3) cells already transformed with pG-KJE8. LB agar and media for cell growth were supplemented with 20 µg/mL chloramphenicol in addition to 100 µg/mL carbenicillin. At inoculation, LB was supplemented with 2 g/L glucose, 100 µg/mL carbenicillin, 20 µg/mL chloramphenicol, and 2 ng/mL anhydrotetracycline per 1 L of media for induction of folding chaperones.

## In vitro enzymatic assembly of lasso peptide BGCs

PURE*frex* 2.1 (Gene Frontier) reactions to express the precursor peptide were assembled according to manufacturer instructions using 1 µL of 200 ng/µL plasmid (pJL1 backbone encoding precursor peptide of interest) per 10 µL reaction and incubated at 37 °C for at least 5 h. Purified proteins were buffer exchanged using Zeba Micro Spin Desalting Columns (7 K MWCO) into synthetase buffer (50 mM Tris-HCl pH 7.5, 125 mM NaCl, 20 mM MgCl₂). 10 µL reactions were then assembled using 5 µL of PURE*frex* reaction, and the appropriate volume of each individual purified enzyme or buffer such that both the RRE and protease were at a final concentration of 10 µM and the cyclase was at a final concentration of 1 µM. Reactions were supplemented to a final concentration of 10 mM DTT and 5 mM ATP and incubated at 37 °C for varying lengths of time. For analysis, samples were desalted using Pierce C18 spin tips (10 µL bed), spotted on a MALDI target plate using 50% saturated CHCA matrix in 80% ACN with 0.1% TFA, and analyzed using a Bruker RapiFlex MALDI-TOF mass spectrometer (flexControl v. 4.0) in reflector positive mode at Northwestern University's Integrated Molecular Structure Education and Research Center (IMSERC). MALDI-TOF data were analyzed using flexAnalysis v. 4.9 (Bruker).

## LC-MS/MS

Reactions were assembled as described above at a scale of 300 µL, desalted using Pierce C18 spin tips, and concentrated to 25 µL using a SpeedVac Vaccuum concentrator system. Samples were then injected on a 1290 Infinity II UHPLC System (Agilent Technologies Inc., Santa Clara, California, USA) onto a Poroshell 120 EC-C18 column (1.9 µm, 50 × 2.1 mm) (Phenomenex, Torrance, California, USA) for C-18 chromatography which was maintained at 45 °C with a constant flow rate at 0.500 ml/min, using a gradient of mobile phase A (water, 0.1 % formic acid) and mobile phase B (100% acetonitrile, 0.1% formic acid). The gradient program was as follows: 0–1 min, 2% B; 1–11 min, 10 – 40% B; 11–12 min, 40–90% B; 12–14 min, hold 90% B; 3 min hold at 10% B. "Targeted MS/MS" in positive ion mode acquisition was conducted on

the samples on an Agilent 6545 quadrupole time-of-flight mass spectrometer equipped with a JetStream ionization source. The source conditions were as follows: Gas Temperature, 325 °C; Drying Gas flow, 13 L/ min; Nebulizer, 35 psi; Sheath Gas Temperature, 275 °C; Sheath Gas Flow, 12 L/ min; VCap, 4000 V; Fragmentor, 175 V; Skimmer, 65 V; and Oct 1 RF, 750 V. The acquisition rate in Auto MS/MS mode was 8 spectra/second, from m/z 100 – 1700 $m/z$ range for MS1 and 3 spectra/second for MS/MS. A ramped collision energy was utilized with a slope and offset of 3.1 and 1, respectively for +2 ions, and a slope and offset of 3.6 and -4.8, respectively for ≥3 ions, and utilizing $m/z$ 121.05087300 and $m/z$ 922.00979800 in positive ion mode as reference masses which is introduced into the ion source by a separate nebulizer and the flow was maintained by an isocratic pump. Additionally, a Targeted Mass Table was created to acquire data on the cyclic peptides (Supplementary Table 5).

### Carboxypeptidase treatment of lasso peptides
Assembled reactions (20 µL scale) were desalted using Pierce C18 spin column and eluted into 20 µL of acetonitrile. After solvent removal under vacuum, reactions were resuspended in a solution containing carboxypeptidase Y at 50 ng/µL in 1X PBS (10 µL) and incubated at room temperature overnight. The mixtures were evaporated to dryness and resuspended in 3 µL saturated α-Cyano-4-hydroxycinnamic acid (CHCA) matrix solution in TFA (trifluoroacetic acid). Samples were then spotted on a matrix assisted laser desorption/ionization (MALDI) plate and analyzed using a Bruker RapiFlex MALDI-TOF mass spectrometer (flexControl v. 4.0) in reflector positive mode at Northwestern University's Integrated Molecular Structure Education and Research Center (IMSERC). MALDI-TOF data were analyzed using flexAnalysis v. 4.9 (Bruker).

### DNA design and preparation for glycosylation
For sequences used in the PglB mutant screen, the wild-type sequence for PglB was retrieved from Uniprot (Q5HTX9) and codon optimized using the IDT Codon Optimization Tool. A codon optimized linker and c-myc tag were appended to the C-terminus of the sequence. Each single variant sequence was then created using the codon optimized wild-type sequence as the template and replacing the respective codon with the most prevalent codon for the replacement amino acid. All protein sequences were ordered as eBlocks with overhang to a linearized pJL1 backbone for use in Gibson Assembly reactions.

The wild-type sequence for *Haemophilus influenzae* protein D was retrieved from Uniprot (Q06282) and codon optimized using the IDT Codon Optimization Tool. Codon optimized linkers, a StrepII tag, and a 6xHis tag were appended to the C-terminus of the PD sequence. Each sequon variant was created by inserting the DNA sequence "AGAGCAGGAGGTGACCAGAACGCTACACGCGCAACCACA" (AA sequence: "RAGGDQNATRATT") between each codon in the wild-type PD sequence. Eleven negative controls were added by instead inserting the DNA sequence "AGAGCAGGAGGTGACCAGTTGGCTACACGCGCAACCACA" (AA sequence: "RAGGDQLATRATT"), in which the asparagine in the sequon is replaced with a leucine that is not glycosylated. All sequon variants were ordered as eBlocks with overhang to linearized pJL1 backbone for use in Gibson Assembly reactions.

The cell-free library generation for the PglB mutant screen was prepared as follows: (1) each backbone gBlock was amplified in a 50 µL PCR reaction with 0.1 ng template added; (2) PCR products were cleaned with a Clean and Concentrate kit (Zymo); (3) cleaned PCR product was diluted to 6.7 ng/µL with nuclease-free water; (4) eBlocks were mixed with their respective pair of backbone gBlocks for a final concentration of 1.5 ng/µL of each component in a 5 µL Gibson reaction; (5) 4 µL of Gibson product was added to a 16 µL rolling circle amplification (RCA) reaction using phi29-XT polymerase (NEB); (6) the completed RCA reaction was diluted 1:1 with the addition of 20 µL nuclease-free water. All PCR reactions used Q5 Hot Start DNA

polymerase (NEB). The diluted RCA product serves as a template for expression of each PglB mutant in CFE. To express sfGFP carrier protein, 200 µL CFE reactions were prepared containing 13.3 ng/µL plasmid encoding the carrier.

The cell-free library generation for the PD sequon walking experiment was prepared using the same workflow as the PglB mutant screen, but with the following exceptions: (1) sequon variant eBlocks and backbone gBlocks were added to 5 µL Gibson reactions for a final concentration of 8 µM of each sequence; (2) Gibson reactions were diluted 6x in nuclease-free water; (3) 1 µL diluted Gibson product was added to 9 µL PCR reactions to generate linear expression templates of each sequon variant. Expression of each sequon variant was performed by adding 1 µL of linear expression template to a 4 µL CFE reaction.

All nucleotide sequences used in the glycosylation portion of this study are provided in the Supplementary Information or in the Supplementary Data 1 File.

### Cell extract preparation
Extract from BL21 Star™ (DE3) cells was prepared based on previous reports[134–136]. Briefly, an overnight culture was used to inoculate a culture of 2 x YTPG at the 10 L scale (target optical density at 600 nm (OD$_{600}$) = 0.06-0.08) in a Sartorius Stedim BIOSTAT Cplus bioreactor. The culture was then incubated at 37 °C with agitation set to 250 RPM. Once the culture reached OD$_{600}$ = 0.6, the cells were induced for T7 RNA polymerase expression by adding IPTG to a final concentration of 0.5 mM. At OD$_{600}$ = 3.0, the cells were harvested and centrifuged at 8,000 x $g$ for 5 min. The resulting cell pellet was then collected and washed 3x with 25 mL of S30 buffer (10 mM Tris acetate pH 8.2, 14 mM magnesium acetate, and 60 mM potassium acetate) by resuspending in cycles of 15 s vortexing and 15 s on ice. In between each wash step, cells were pelleted via centrifugation at 10,000 x $g$ for 2 min and the supernatant was poured off. After the final wash step, the supernatant was poured off, the mass of the cell pellet was recorded, and the cell pellets were flash frozen and stored at -80 °C. For lysate preparation, the cell pellets were thawed on ice for 1 hr. Next, 1 mL of S30 buffer per gram of cell pellet was added to each tube. The cells were then resuspended via vortexing, again in cycles of 15 s vortexing and 15 s on ice. After resuspension, the cells were then lysed via homogenization using a single pass through an Avestin EmulsiFlex-B15 homogenizer between 20,000–25,000 psig. Following homogenization, the lysed sample was centrifuged at 12,000 x $g$ for 10 min at 4 °C. The supernatant was then collected and centrifuged again at 12,000 x $g$ for 10 min at 4 °C. Following the final centrifugation, the supernatant was pooled, aliquoted, flash frozen, and stored at -80 °C until use.

For extracts enriched with C*j*PglB$^{Q287K}$ and CPS from *S. pneumoniae* serotype 4 and derived from Hobby strain[105], the above directions provided for BL21 Star™ (DE3) cells were followed with the following changes: Prior to growing the overnight cultures, electrocompetent Hobby cells were transformed with pSF-C*j*PglB$^{Q287K}$-LpxE-KanR and pB-4[104] and plated on LB agar plates containing 50 mg/mL Kanamycin and 20 mg/mL of tetracycline. During each cell growth phase, the cultures were also supplemented with 50 mg/mL Kanamycin and 20 µg/mL of tetracycline. At OD$_{600}$ = 0.6-0.8, the culture was supplemented with 0.1% w/v arabinose in addition to 0.5 mM IPTG to induce for C*j*PglB$^{Q287K}$ and CPS4 expression respectively, and the incubator was turned down to 220 RPM and 30 °C. Additionally, the supernatant from the first 12,000 x $g$ centrifugation spin was collected and underwent runoff by wrapping the tubes in aluminum foil and incubating at 37 °C and 250 RPM for 1 h. Following runoff, the tubes were centrifuged at 10,000 x $g$ at 4 °C for 10 min and the supernatant was collected, mixed, and aliquoted before flash freezing and storing at -80 °C until use.

### Crude membrane fraction
For producing crude membrane fraction, Hobby strain cells were transformed with pB-4[104] and plated on LB agar plates containing

20 µg/mL of tetracycline. A single colony was then used to inoculate a 50 mL overnight culture of LB supplemented with 20 µg/mL of tetracycline. The next morning, 1 L of 2xYTPG supplemented with 20 µg/mL of tetracycline was inoculated with a target starting $OD_{600}$ = 0.06-0.08. The culture was the incubated at 37 °C with agitation set to 250 RPM. At $OD_{600}$ = 0.6-0.8, the culture was supplemented with 0.5 mM IPTG and the culture was then incubated overnight at 30 °C and agitation of 220 RPM. The next morning, the cells were harvested via centrifugation at 8,000 x $g$ for 5 min at 4 °C. After pouring off the supernatant, 1 mL per gram of cell pellet of resuspension buffer (50 mM Tris HCl, pH 7.5, 25 mM NaCl) was added to the pellets. The cells were then resuspended via vortexing in cycles of 15 s vortexing and 15 s on ice. After the cells were fully resuspended, the sample was lysed via homogenization using a single pass through an Avestin EmulsiFlex-B15 homogenizer between 20,000–25,000 psig. Following lysis, the sample was then centrifuged at 12,000 x $g$ at 4 °C for 30 min. The supernatant was then ultracentrifuged at 100,000 x $g$ at 4 °C for 1 h to pellet the membrane vesicles. Following ultracentrifugation, the supernatant was poured off and 0.2 mL/gram original cell pellet of resuspension buffer (50 mM Tris HCl, pH 7.5, 25 mM NaCl, 1% w/v DDM) was added to the pellet before incubating overnight at 4 °C on a shake table. The next morning, the samples were pipette mixed to ensure complete resuspension of the pellet and the samples were incubated at room temperature for 30 min. Finally, the samples were spun at 16,000 x $g$ for 1 h at 4 °C and the supernatant was mixed, aliquoted, flash frozen, and stored at -80 °C until use.

### Nanodisc supplemented CFE reactions

In vitro expression of each WT or mutant PglB construct was performed by adding 1 µL of diluted RCA product to a 4 µL BL21 Star™ (DE3) CFE reaction supplemented with 66.7 µM MSP1E3D1 POPC nanodiscs (Cube Biotech). CFE reactions were carried out using the PANOx-SP reaction, with reaction formulations previously described[45,137,138]. Briefly, reactions were assembled with the following final concentrations: 8 mM magnesium glutamate, 10 mM ammonium glutamate, 130 mM potassium glutamate, 1.2 mM ATP, 0.85 mM GTP, 0.85 mM UTP, 0.85 mM CTP, 34 µg/mL folinic acid, 0.17 mg/mL tRNA, 0.4 mM nicotinamide adenine dinucleotide (NAD), 0.27 mM coenzyme A (CoA), 4 mM oxalic acid, 1 mM putrescine, 1.5 mM spermidine, 57 mM HEPES pH 7.2, 2 mM of each of the 20 standard amino acids, 33 mM phosphoenolpyruvate (PEP) and 30% v/v cell extract. All reactions were incubated at 30 °C overnight.

### In vitro glycosylation reactions

In the PglB mutagenesis screen, in vitro glycosylation reactions were performed by combining 0.4 µL unpurified sfGFP acceptor, 1 µL unpurified PglB mutant, and 3 µL *S. pneumoniae* CPS4 crude membrane fraction in a 5 µL reaction volume containing 0.1% w/v DDM (Anatrace), 1% w/v Ficoll 400 (Sigma), 10 mM manganese chloride (Sigma), and 50 mM HEPES (Sigma).

In the sequon walking experiment, in vitro glycosylation reactions were performed by combining 1 µL unpurified sequon variant and 2.5 µL enriched extract containing *Cj*PglB$^{Q287K}$ and *S. pneumoniae* CPS4 in a 5 µL reaction volume containing 0.1% w/v DDM, 1% w/v Ficoll 400, 10 mM manganese chloride, and 50 mM HEPES.

### AlphaLISA reactions for glycoconjugates

Completed in vitro glycosylation reactions were diluted in a buffer consisting of 50 mM HEPES pH 7.4, 150 mM NaCl, 1 mg/mL BSA, and 0.015% v/v Triton X-100. All glycoconjugate AlphaLISA experiments were performed with 1 µL reaction volumes with a 0.08 mg/mL final concentration of Protein A donor beads and 0.02 mg/mL final concentration of anti-6xHis acceptor beads, which immobilize the *S. pneumoniae* CPS4 antiserum and the 6xHis-tagged glycoconjugates, respectively. Following dilution,

an Echo 525 acoustic liquid handler was used to dispense 0.25 µL diluted in vitro glycosylation product, 0.25 µL *S. pneumoniae* CPS4 antiserum, 0.25 µL blank buffer, and 0.125 µL anti-6xHis acceptor beads diluted in buffer from a 384-well polypropylene 2.0 Plus Source microplate using the 384PP_Plus_GPSA fluid type into an AlphaPlate 1536-well destination microplate (Revvity). The plate was sealed and equilibrated for 1 h at room temperature. Following incubation, 0.125 µL of Protein A donor beads diluted in buffer were transferred to each reaction. Reactions were equilibrated for an additional hour at room temperature in the dark. For analysis, reactions were incubated for 10 min in a Biotek Synergy Neo2 plate reader at room temperature, and chemiluminescent signal was read using the AlphaLISA filter with an excitation time of 100 ms, an integration time of 300 ms, and a settle time of 20 ms. For the PD sequon walking experiment, replicate AlphaLISA reactions were performed on separate plates. Signal for each plate was normalized using the formula:
$$Normalized\ signal = \frac{(Raw\ signal - Mean\ neg.control\ signal)}{Mean\ pos.control\ signal}.$$
Results were visualized using Prism version 10.3.1 (GraphPad).

### Western blotting

Samples were loaded on a 4–12% Bis-Tris gel and run with either MOPS SDS or MES SDS buffer for 45 min at 200 V. A semidry transfer cell was then used to transfer the samples to Immobilon-P-poly(vinylidene difluoride) PVDF 0.45 µm membranes at 80 mA per blot for 45 min. After transferring, the membranes were blocked for 30 min at room temperature in Intercept Blocking Buffer (Licor) with gentle shaking. Following blocking, the blots were briefly rinsed with 1x PBST and then probed for 1 h at room temperature with gentle shaking using one of the following antibodies diluted into Intercept Blocking Buffer with 0.2% Tween20: anti-6xHis (Abcam, ab1187) at 1:7500 dilution, type 4 pneumococcal antiserum (Cedarlane, 16747(SS)) at 1:1000 dilution, or anti-myc (Abcam, ab9106) at 1:1000 dilution. Following primary incubation, membranes were rinsed twice with 1x PBST followed by 3 five min washes in 1x PBST at room temperature with gentle shaking. Following washing, the blots were probed for 1 h at room temperature with gentle shaking using a fluorescent goat, anti-rabbit antibody GAR-680RD (Licor, 926-68071) at a dilution of 1:10,000 in Intercept Blocking Buffer with 0.2% Tween20 and 0.1% SDS. Then, the membranes were washed as described earlier. Finally, the blots were imaged with either a Licor Odyssey Fc or an Azure 600 imager and analyzed by densitometry using Licor Image Studio Lite (v. 5.2.5). The fluorescence background was subtracted from each membrane before assessing densitometry. All uncropped dot blots and Western blots are provided (Supplementary Figs. 27–33).

### Statistics and reproducibility

All sample sizes, error bars, and statistical tests are defined in Figure legends. No statistical method was used to predetermine sample size. No data were excluded from the analyses. The experiments were not randomized, and researchers were not blinded to the experimental conditions. Statistical analyses were performed using Excel version 16.95.4 and GraphPad Prism version 10.3.1.

### Reporting summary

Further information on research design is available in the Nature Portfolio Reporting Summary linked to this article.

## Data availability

Source data are provided with this paper (data split between four source data files, with the relevant file noted in each figure caption). All sequences and accession codes for proteins used throughout this study are included in the Supplementary Information or in

Supplementary Data 1. Protein structures used in this work include a homology model of *C. jejuni* PglB[94] and PDB ID: 8CWP. The LC-MS/MS data generated in this study has been deposited in the Zenodo repository under https://doi.org/10.5281/zenodo.15385022. Source data are provided with this paper.

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

## Acknowledgements

The authors would like to thank Rui Gan, Jonathan Bogart, and Thuy Aziz for helpful discussions. This work was supported by the National Institutes of Health (NIH) 1U19AI142780-01 (R.N., E.P.B., and M.C.J.), DTRA (HDTRA1–20-1-0004) (M.C.J.), the National Science Foundation (CBET - 1936789) (M.C.J.), and DARPA (W911NF-23-2-0039) (M.P.D., A.S.K., and M.C.J.). D.A.W. acknowledges support from the National Science Foundation Graduate Research Fellowship under grant number DGE-1842165. Z.M.S. acknowledges support from the National Science Foundation National Research Traineeship under grant number 2021900. M.D. acknowledges support from the Canadian Institutes of Health Research Postdoctoral Fellowship under grant no. MFE-176575. This work made use of the IMSERC MS facility at Northwestern University, which has received support from the Soft and Hybrid Nanotechnology Experimental (SHyNE) Resource (NSF ECCS-2025633), the State of Illinois, and the International Institute for Nanotechnology (IIN).

## Author contributions

D.A.W. designed research, performed experiments, performed MALDI-MS on reactions, analyzed data, and wrote the paper. Z.M.S. designed research, performed experiments, analyzed data, and wrote the paper. M.D.C. designed research, performed experiments, performed MALDI-MS on reactions, analyzed data, and edited the paper. M.D. computationally identified all lasso peptide BGCs and wrote the paper. K.F.W. performed experiments. D.V.P. performed experiments. S.E.S. performed experiments. R.F. performed experiments. F.T. optimized liquid chromatography and mass spectrometry parameters. S.K.F. analyzed proteomics data. S.W.H. designed research. P.F. supervised research. R.N. supervised research and edited the paper. M.P.D. supervised research and edited the paper. E.P.B. supervised research and edited the paper. A.S.K. supervised research, analyzed data, and edited the paper. M.C.J. designed and directed research, analyzed data, and wrote the paper.

## Competing interests

M.C.J. and M.P.D. have a financial interest in National Resilience and Gauntlet Bio. M.C.J. also has a financial interest in Stemloop Inc. and Synolo Therapeutics. M.C.J.'s interests are reviewed and managed by Northwestern University and Stanford University in accordance with their competing interest policies. M.P.D.s interests are reviewed and managed by Cornell University. All other authors declare no competing interests.

## Additional information

[1]Department of Chemical and Biological Engineering, Northwestern University, Evanston, IL 60208, USA. [2]Chemistry of Life Processes Institute, Northwestern University, Evanston, IL 60208, USA. [3]Center for Synthetic Biology, Northwestern University, Evanston, IL 60208, USA. [4]Interdisciplinary Biological Sciences Program, Northwestern University, Evanston, IL 60208, USA. [5]Medical Scientist Training Program, Northwestern University, Evanston, IL 60208, USA. [6]Broad Institute of MIT and Harvard, Cambridge, MA 02142, USA. [7]Department of Chemistry and Chemical Biology, Harvard University, Cambridge, MA 02138, USA. [8]Department of Chemistry, Northwestern University, Evanston, IL 60208, USA. [9]Integrated Molecular Structure Education and Research Center (IMSERC), Northwestern University, Evanston, IL 60208, USA. [10]Proteomics Center of Excellence, Northwestern University, Chicago, IL 60611, USA. [11]Biochemistry, Molecular and Cell Biology (BMCB) Program, Cornell University, Ithaca, NY 14853, USA. [12]Robert Frederick Smith School of Chemical and Biomolecular Engineering, Cornell University, Ithaca, NY 14853, USA. [13]Cornell Institute of Biotechnology, Cornell University, Ithaca, NY 14853, USA. [14]Howard Hughes Medical Institute, Harvard University, Cambridge, MA 02138, USA. [15]Department of Bioengineering, Stanford University, Stanford, CA 94305, USA. [16]These authors contributed equally: Derek A. Wong, Zachary M. Shaver. ✉e-mail: balskus@chemistry.harvard.edu; ashty.karim@northwestern.edu; mjewett@-stanford.edu

