## [Peer Review file · Nature Communications]

Characterizing and engineering post-translational modifications with high-throughput cell-free expression

Corresponding Author: Professor Michael Jewett

Version 0:

Reviewer comments:

Reviewer #1

(Remarks to the Author)

The manuscript from the Jewett, Karim, and Blaskus groups describe a novel methodology for the rapid detection and functional demarcation of the interactions that drive the recognition of substrate peptides by their cognate binding domains in the synthesis of RiPP classes of natural products. The moderately high-throughput nature of the platform allows for rapid elucidation of binding and determination of key residues in such interactions across multiple classes of RiPP pathways. The authors then leverage this platform for the identification of a new lasso-peptide that was identified through genome mining, to show proof of concept of this approach. Given the nature of the study and the successful demonstration of discovery, the work would be of broad interest to the readership to Nature Comm. However, there are a few critical control experiments and several editorial changes in the manuscript that are requested before the current work can be considered suitable for publication.

1. The identification of alphasassin as a lasso peptide is not particularly convincing. The authors use carboxypeptidase digestion as a proxy in claiming that the product is a lasso but this is not convincing. Looking at SI Figure 9, clearly the modified peptide also undergoes some digestion. Ideally, the authors could carry out NMR analysis of the product. However, if compound availability is limiting, other experiments (such as tandem MS analysis of the digested, modified peptide, or heat stability of the product) could be added to further bolster the claim that alphasassin is a lasso.
2. An important control for the alphasassin reconstitution work would be to add a cyclase from an orthologous pathway to show that lasso formation is specific only to the AlphaB1, AlphaB2 and AlphaC combination. Presumably, substitution of a different cyclase should produce only the leader-cleaved (but uncyclized) product.

Minor points:

1. Line 51: not all RiPPs are composed of an amino acid backbone (i.e. thiopeptides, thioviridamides, etc.). Please rephrase.
2. Line 62 "in over 50% of RiPP BGCs..." The referenced publication is nearly a decade old and far more RiPP BGCs that lack an RRE have since been identified since. Again, please consider rephrasing.
3. Please include protein gels to support the data presented in line 287-294. This can be done as an SI figure similar to SI Figure 1.
4. Line 117: TbiB1 comes from a putative lasso cluster as no final product from this pathway has been formally identified. Please revise text here and in Fig 1 legend accordingly.

Reviewer #2

(Remarks to the Author)

In this manuscript from the Blaskus, Karim, and Jewett labs, the authors describe a high-throughput workflow to characterise the molecular interactions that permit RiPP biosynthesis to occur, namely those between the RiPP precursor peptide and the so-called RRE domain (RiPP recognition element). The strength of this paper is the novelty of the AlphaLISA method and its enhanced throughput. I find this first section of the paper to be remarkable and creative; the justification for the need of

this method is also well-stated in the paper. I suspect many researchers will want to adopt the methods disclosed within. However, there are several major weaknesses in this paper that must be addressed before publication in any journal, let alone a premier journal like Nature Communications. Among the major concerns are the omissions in the cited references and the authors claiming they discovered something new that has already been reported. Further, the authors describe their method as solving a major obstacle that honestly doesn't represent an actual obstacle. Unfortunately, trying to solve a problem that doesn't exist comes across as misguided or uninformed. Taken together, there are simply too many conceptual errors, misrepresentations, missing citations/prior work, etc. to recommend revision thus I recommend rejection. I hope my well-intended comments below are helpful to the authors as they work to improve this manuscript.

MAJOR ISSUES:

Line 115: I am confused why the authors refer to thiomuracin as a computationally predicted thiopeptide. This suggests they are unaware of the work from at least five research groups (Bowers, Mitchell, Nair, Suga, van der Donk) that have published numerous papers on this pathway.

Lines 172-174: "we next assessed if our platform could accurately determine residues important for binding within the leader sequence of a previously discovered, but uncharacterized precursor peptide." This statement suggests the authors are unaware of the work mentioned in the previous comment. A considerable amount of characterization has been performed on TbtA -- this entire section simply replicates the work of others performed about 5-10 years ago and unfortunately, it is presented as new information.

Figure 2: Why are the authors including "SRRGS" at the beginning of the TbtA peptide? This portion is clearly a leftover from a cloning or cleavage step. Of course, mutation of these residues to Ala will have no contribution, they are not part of the native peptide! TbtA starts with Met. I am worried that the authors are using these residues improperly in other % identity calculations. Stating that the peptide 38% TbtA sequence also seems to be misleading to me; isn't this peptide 17/24 the same (71% TbtA)?

Line 225: This statement about RREs being incorrectly identified as a challenge for RiPP discovery is not true. There is a reliable, HHpred-based tool for RRE detection that three groups developed a number of years back, Medema, van Wezel, Mitchell, which is now quite popular. Also, as the authors point out, many are readily detected as homologs of PqqD which is a simple BLAST search. I have never heard of a single case of a misidentified RRE being a problem in RiPP discovery.

Line 227: The authors here claim they are working on an uncharacterized lasso peptide predicted by AntiSMASH. As a matter of fact, this lasso peptide was bioinformatically reported in 2017 (Nat Chem Biol, Mitchell) and it was experimentally characterized in 2023 (referred to as LP-1010) (Nat Microbiol, Voigt).

Line 238: It is odd that the authors use antiSMASH to construct a much smaller dataset than those published by other groups. At minimum, it should be acknowledged that their dataset is not "a complete collection", as it considerably smaller than other published datasets.

Line 297: As stated above, Alphasassin has previously been characterized by the Voigt group (referred to as LP-1010). Does this peptide have antibacterial activity? How much was isolated? Why stress the cationic features and not do these experiments?

MINOR ISSUES:

Although most journals do not require it, it is advisable if all authors would include accession identifiers for all proteins discussed in the main text of their manuscripts. Biochemistry requires this and Springer Nature should too. This helps literature mining algorithms that link protein sequences with experimentally determined functional data.

In the RiPP field, it is customary to use negative numbering for the leader peptide. The authors should follow the convention established by the RiPP community. And please do not include residues that are not part of the native peptide.

Line 66: PqqD is not an "enzyme". It is a PqqA-binding domain. The enzyme formally is PqqE. See work from Klinman and Latham.

Line 121: For the sake of clarity, the term PqqD should only be used when referring to bona fide PqqD. When referring to an RRE from some other RiPP pathway, "PqqD" should be explicitly avoided. Use RRE or the name of the protein, if it has one.

Line 150: odd to call this pathway a derivative of GE2270, when the pathway encodes thiomuracin, a known thiopeptide with many supporting publications.

Lines 155-170: the AlphaLISA method appears strong regarding throughput, but it appears to only be qualitative and somewhat lacking in detecting small differences in binding strengths. The authors seem to minimize the effectiveness of other assays (like ITC, NMR or FP), despite their lower throughput nature. One must admit the data gleaned from these older methods is more quantitative and sensitivity to binding changes is readily detected. I don't see that as being the case for AlphaLISA. Perhaps these two assays will become complementary, with AlphaLISA used as a first-pass rather than it supplanting other methods, as the narrative currently implies. It is important to balance the pros and cons of any new method, not just sell the pros.

Lines 199-200. This is the mean of 3 technical replicates, but what is the standard deviation? Do we get a Kd out of these data? How reproducible are these data upon biological replication? Prep-to-prep variability is often an issue, for instance. For establishment of a new method, this is critical data to include.

Lines 245 and 247: the term "pro-peptide" is not encouraged, see 2013 Arnison et al community review on RiPP guidelines and nomenclature.

Line 248: This seems like a bait and switch. Are the authors trying to find lasso peptides that act as an antibiotic? If so, where are the antibacterial assays?

Figure 3: Which examples had multiple RREs and how was this determined? Did RRE-Finder confirm these predictions? The authors should use the best tools available, not older insufficient tools that make their results sound more appealing. Many of these data are from a single replicate, thus without replication should not be published. Others have five replicates. I am confused by the lack of consistency and lack of rational.

Line 311: a MALDI instrument typically cannot give this much accuracy for a mass measurement. Do not report non-significant figures.

Line 316: Notwithstanding the former comment, a MALDI instrument should not be off by 1 Da either. Something is wrong here. The error is most likely in the tenths place (not the ones place nor the ten-thousands place, per last comment).

Lines 360 to 362: I have never heard of a case where the misannotation of RiPP biosynthetic enzymes was the offending issue. Poor expression/solubility of the biosynthetic enzymes in a heterologous host, however, is a notorious problem in RiPP biosynthesis and should be considered here.

Lines 366-368: This statement is an oversimplification and problematic for several reasons: Just because an RRE is active does not mean the rest of the enzymes will be active. As shown by Klinman, Nair, Naismith, Severinov, van der Donk, Mitchell, Piel, Hegemann, Schulman, and a great number of additional other labs, RREs are easy to express and purify in functional form. Other RiPP biosynthetic components are typically much less well behaved and thus it is misleading to state the claim the authors are making. Conversely, just because an RRE failed to express in the PurExpress setting (which lacks chaperones, crowding effects, and the general robustness of a living cell) does not mean it would not work in a heterologous expression system. There is a large amount of literature to support the opposite of what the authors state here. Lastly, I am uncertain of how the 40% figure was calculated.

The SI would be a good place to enhance rigor and data transparency for the lasso peptide section. Showing Coomassie-stained SDS-PAGE gels as well for every protein used in the study is a must. Accession codes, or at the very least the amino acid sequence, should be given for all of the lasso peptide precursor peptides.

Reviewer #3

(Remarks to the Author)

In this manuscript, Wong et al describe a system that has the potential to improve the speed and scale of RRE/precursor peptide identification. Ribosomal natural products (RiPPs) are produced through the expression of a precursor peptide that gets post-translationally modified by one or more enzymes. Most RiPPs discovered to date possess a leader peptide, which is important for recognition by parts of the biosynthetic machinery. In about 50% of RiPP biosynthetic gene clusters one finds one or more RiPP recognition element(s) (RREs), which are responsible for binding to the leader peptide. The presented study uses a cell-free expression system to produce RRE/precursor peptide pairs and an in-solution ELISA to probe the interaction. I found the manuscript very well written and the method presented exciting. In my mind, the study is going to attract a lot of attention amongst RiPP researchers and provides an interesting starting point for other fields. I therefore recommend publication in Nat Commun after the following issues have been addressed:

1. The authors express the RREs as MBP fusion proteins, presumably because the RREs cannot be expressed without this solubility tag. This results in two issues:

- a) The authors should include a negative control using just MBP, without an RRE, to demonstrate that the interaction is specific.
- b) The authors should characterize several of the RREs once they have been cleaved off MBP. It is a well known problem that MBP is powerful enough to keep misfolded and aggregated proteins in solution, and the expression in Supplementary Figure 1 does not look very strong, another indicator that the RRE may be "unhappy".

2. The authors characterize a large number of RRE/precursor peptide pairs from lasso peptide clusters identified via genome mining. While in principle a great way of probing many interactions in parallel, I could not find data demonstrating expression of all the RRE-MBP fusions employed. Are the non-binders actually not binding, or the RRE simply not expressed? The authors should show at least a gel for the RRE-MBP fusions of figure 3.

3. The authors claim to have made a new lasso peptide. To make that claim requires the authors to isolate that products, or at least detect it, in the natural producer. Otherwise the wording should be amended to assert the production of a peptide

with the topology of a lasso peptide. Which brings me to the analytics of the new lasso peptide: While treatment with carboxypeptidase Y is a start, more experiments are required to assert the proposed structure/topology is correct. Don't worry, I understand that an NMR would be unreasonable, but MS fragmentation data could easily be acquired with the small quantities produced.

Version 1:

Reviewer comments:

Reviewer #1

(Remarks to the Author)

The authors have addressed most all of the concerns raised during the initial review of this manuscript. I would suggest addition of a few more references (especially with regards to work on the thiipeptide system) for the sake of completeness. Otherwise, this is fine for publication.

Reviewer #2

(Remarks to the Author)

The authors did a tremendous job in responding to the prior critique. The current manuscript removed all of the concerning issues and even added a new dimension of data with glycosylation. I believe the manuscript is ready for publication.

Reviewer #4

(Remarks to the Author)

In the present manuscript Wong and colleagues describe their design of a cell-free expression system, which is high throughput and adaptable for screening large libraries of carrier protein and OST mutants. Moreover, the system is adaptable to study posttranslational modifications, including protein glycosylation. As my expertise is on glycobiology, I have focused my review on this aspect of the work.

The authors make quite strong claims on their workflow being suitable for vaccine design. Although I agree that their workflow is suitable for accelerated design and perhaps highthroughput screening of vaccines, in my opinion the actual proof of their concept is lacking. For vaccines to work the product needs to be as pure as possible. Yet, the authors generate a variety of products as can be seen in the western blots of supplementary figures 18 and 21. The authors claim these are longer CPS fragments, but as they used a "crude membrane fraction" these products could indeed be longer CPS variants, but the addition of completely other or multiple polysaccharide fragments is a real possibility. The anti-CPS4 anti-serum used for instance in figure 4, proves that CPS4 is present, but doesn't prove that no other products are also included in their mixture. Therefore, to fully verify their claims the authors should include a mass spectrometry glycoproteomic analysis, including at least the different products they get in the glycoprotein sfGFP and CPS4 reaction.

Moreover, also the non-glycosylated AQNAT has an additional product, which seems enriched in the DQNAT. What is this additional band?

Version 2:

Reviewer comments:

Reviewer #4

(Remarks to the Author)

I thank the authors for the additional work and for clarifying the issue I raised in my report.

Note to reviewers:

We appreciate the review of our manuscript, enthusiasm for the methods, and request for revision. Our revised manuscript has addressed the concerns raised, including additional clarifying control experiments and contextualization of our work around the discovery of the final described lasso peptide. Beyond these changes, there was interest in further application of our workflow. To address this interest, enhance our manuscript, and demonstrate the generalizability of our workflow, we explored whether a modified version of our platform could be used to engineer other post-translational modification (PTM) systems important in medicine. To do this, we applied the method to understand and engineer glycosylation of biologics (e.g., conjugate vaccines; new Figures 4-6). Specifically, we designed, expressed, and assayed hundreds of oligosaccharyltransferases for their ability to transfer a clinically relevant glycan onto a carrier protein, identifying a mutant with 1.7-fold higher activity. Using this mutant, we then scanned 328 locations on an FDA-approved carrier protein for sites permissible to *in vitro* glycosylation, which can be used towards designing vaccines with improved immunogenicity. By applying our methodology to two distinct classes of PTMs, we have demonstrated how our workflow can accelerate design-build-test cycles for biologics with complex PTMs and improved therapeutic properties. In light of this new application, we have recentered the manuscript focus on the methodology itself, which we believe better showcases the utility of our approach and potential applications in both natural products and vaccine development research.

Point-by-point response.**Reviewer #1 (Remarks to the Author):**

The manuscript from the Jewett, Karim, and Balskus groups describe a novel methodology for the rapid detection and functional demarcation of the interactions that drive the recognition of substrate peptides by their cognate binding domains in the synthesis of RiPP classes of natural products. The moderately high-throughput nature of the platform allows for rapid elucidation of binding and determination of key residues in such interactions across multiple classes of RiPP pathways. The authors then leverage this platform for the identification of a new lasso-peptide that was identified through genome mining, to show proof of concept of this approach. Given the nature of the study and the successful demonstration of discovery, the work would be of broad interest to the readership to Nature Comm. However, there are a few critical control experiments and several editorial changes in the manuscript that are requested before the current work can be considered suitable for publication.

Thank you for your accurate summary and recognition of the broad interest of our work.

1. The identification of alphasassin as a lasso peptide is not particularly convincing. The authors use carboxypeptidase digestion as a proxy in claiming that the product is a lasso but this is not convincing. Looking at SI Figure 9, clearly the modified peptide also undergoes some digestion. Ideally, the authors could carry out NMR analysis of the product. However, if compound availability is limiting, other experiments (such as tandem MS analysis of the digested, modified peptide, or heat stability of the product) could be added to further bolster the claim that alphasassin is a lasso.

Thank you for the opportunity to provide additional evidence that alphasassin (now Las24/Las-1010; see response to reviewer 2 below) is produced. We scaled-up reactions producing Las24 and performed LC-MS/MS to confirm the identity and structure of our product. We have

provided the spectra from this experiment in **Supplementary Figure 14** and have referenced this figure in the main text of the manuscript on p. 11.

*“Subsequent characterization experiments confirmed that the production of this lasso peptide, Las24, is time dependent (**Supplementary Fig. 12**), that each of the proteins in the predicted BGC is necessary for maturation (**Supplementary Fig. 13**), that the sequence of the molecule matches the expected structure (**Supplementary Fig. 14**), that the molecule is resistant to carboxypeptidase (a common confirmation of threaded topology) (**Supplementary Fig. 15**), and that there is limited to no interaction of Las24 biosynthetic components with those from other lasso peptide BGCs (**Supplementary Fig. 16**).”*

2. An important control for the alphasassin reconstitution work would be to add a cyclase from a orthologous pathway to show that lasso formation is specific only to for the AlphaB1, AlphaB2 and AlphaC combination. Presumably, substitution of a different cyclase should produce only the leader-cleaved (but uncyclized) product.

We thank you for this suggestion and agree that confirming the specificity of Las24 (Las-1010) biosynthetic components would further strengthen our characterization of the BGC. To add this control, we purified the tailoring enzymes from another lasso peptide BGC (Fuscanodin/Fusilassin^{1,2}) and performed an additional experiment mixing various combinations of Las24 and Fuscanodin BGC components. Only in reactions containing Las24 BGC components alone, did we see robust production of the mature lasso peptide. We have included spectra from this experiment in **Supplementary Figure 16** and have also referenced this in the manuscript on p. 11 as described in the previous comment.

Minor points:

1. Line 51: not all RiPPs are composed of an amino acid backbone (i.e. thiopeptides, thioviridamides, etc.). Please rephrase.

We have amended the mentioned sentence to clarify that although mature RiPPs may not be composed of only amino acids, the initial precursor peptide is composed of all amino acids. The revised sentence on p. 4 now reads as:

“While mature RiPPs vary in amino acid composition, RiPPs originate as a precursor peptide typically composed of an N-terminal leader sequence and C-terminal core sequence”

2. Line 62 "in over 50% of RiPP BGCs..." The referenced publication is nearly a decade old and far more RiPP BGCs that lack an RRE have since been identified since. Again, please consider rephrasing.

Thank you for this point. A more recent publication in 2020 found that of the 29 classes of RiPP produced by prokaryotes, RRE's are found in 19 of them³. We have revised the sentence on p. 4 to reflect this finding and have also incorporated the additional citation.

“In around 65% of RiPP classes produced in prokaryotes, the recognition of the leader sequence by tailoring enzymes is facilitated by a standalone protein or portion of a fusion protein containing a RiPP precursor peptide recognition element (RRE)³.”

3. Please include protein gels to support the data presented in line 287-294. This can be done as an SI figure similar to SI Figure 1.

We have updated the supplementary figures to include additional protein gels to support the data. **Supplementary Figure 8** demonstrates soluble expression in PURE_{flex} of all computationally predicted lasso peptide RREs tested in this study. **Supplementary Figure 10** shows a Coomassie SDS-PAGE gel of purified tailoring enzymes used to reconstitute the Las24 (Las-1010) BGC. **Supplementary Figure 17** shows a Coomassie SDS-PAGE gel of purified tailoring enzymes from the Fuscanodin/Fusilassin BGC.

4. Line 117: TbiB1 comes from a putative lasso cluster as no final product from this pathway has been formally identified. Please revise text here and in Fig 1 legend accordingly.

We thank you for this point and have adjusted the text accordingly. With the reframing of the manuscript, the original line 117 has been removed, and the caption of Fig 1. on p. 5 has been revised to:

“(c) a putative lasso peptide from Thermobaculum terrenum ATCC BAA-798”

Reviewer #2 (Remarks to the Author):

In this manuscript from the Balskus, Karim, and Jewett labs, the authors describe a high-throughput workflow to characterise the molecular interactions that permit RiPP biosynthesis to occur, namely those between the RiPP precursor peptide and the so-called RRE domain (RiPP recognition element). The strength of this paper is the novelty of the AlphaLISA method and its enhanced throughput. I find this first section of the paper to be remarkable and creative; the justification for the need of this method is also well-stated in the paper. I suspect many researchers will want to adopt the methods disclosed within.

We appreciate your recognition of the utility and broad interest of our methods.

However, there are several major weaknesses in this paper that must be addressed before publication in any journal, let alone a premier journal like Nature Communications. Among the major concerns are the omissions in the cited references and the authors claiming they discovered something new that has already been reported. Further, the authors describe their method as solving a major obstacle that honestly doesn't represent an actual obstacle. Unfortunately, trying to solve a problem that doesn't exist comes across as misguided or uninformed. Taken together, there are simply too many conceptual errors, misrepresentations, missing citations/prior work, etc. to recommend revision thus I recommend rejection. I hope my well-intended comments below are helpful to the authors as they work to improve this manuscript.

Thank you for your comments and important concerns about our work. Our intention with submitting this study was to share our findings as they occurred throughout the project with integrity. We regret missing an essential publication that came out mid-project that changes the described novelty of the RiPP that we produced. We have seriously and thoroughly revised our manuscript, recontextualizing it around the previously discovered Las1010. Inspired by your comments above about our method's novelty, we also chose to recenter the manuscript's focus on the methodology itself by showing application of the workflow for the study of additional PTMs, namely protein glycosylation. Our hope is that by both revising our initial study and by adding additional use cases in protein engineering and glycosylation that we have addressed your concerns and significantly improved the manuscript for publication in *Nature Communications*.

MAJOR ISSUES:

Line 115: I am confused why the authors refer to thiomuracin as a computationally predicted thiopeptide. This suggests they are unaware of the work from at least five research groups (Bowers, Mitchell, Nair, Suga, van der Donk) that have published numerous papers on this pathway.

This was an oversight, and we have updated the text with appropriate citations. Specifically, we have incorporated the citations listed below:

Hudson, G. A., Zhang, Z., Tietz, J. I., Mitchell, D. A. & van der Donk, W. A. In Vitro Biosynthesis of the Core Scaffold of the Thiopeptide Thiomuracin. *J Am Chem Soc* **137**, 16012-16015 (2015). <https://doi.org/10.1021/jacs.5b10194>

Zhang, Z. *et al.* Biosynthetic Timing and Substrate Specificity for the Thiopeptide Thiomuracin. *Journal of the American Chemical Society* **138**, 15511-15514 (2016). <https://doi.org/10.1021/jacs.6b08987>

These citations have been incorporated into the main text of the manuscript on p. 5-6 within the following text:

“To do this, we chose the RRE domain of TbtF, the cyclodehydratase involved in thiomuracin^{4,5} biosynthesis, and its leader sequence of TbtA (Fig. 2a). Mutating residues L(-32), L(-29), M(-27), D(-26), and F(-24) within the leader sequence of TbtA to an alanine was previously shown using fluorescence polarization to have a reduction in binding affinity to TbtF⁵.”

Lines 172-174: "we next assessed if our platform could accurately determine residues important for binding within the leader sequence of a previously discovered, but uncharacterized precursor peptide." This statement suggests the authors are unaware of the work mentioned in the previous comment. A considerable amount of characterization has been performed on TbtA -- this entire section simply replicates the work of others performed about 5-10 years ago and unfortunately, it is presented as new information.

Thank you for pointing this out. We have reworked the main text to directly compare our results with those detailed in Zhang *et al*⁵. In short, our results with TbtA correlate strongly with data obtained using fluorescence polarization, which strengthens our argument that our method is robust and can be used to complement traditional assays. Additionally, our ability to quickly construct and test peptide libraries (**Figure 2b**) enables us to rapidly confirm the importance of specific residues identified using the alanine scan data as well as obtain in-depth design rules faster than were previously determined.

“Mutating residues L(-32), L(-29), M(-27), D(-26), and F(-24) within the leader sequence of TbtA to an alanine was previously shown using fluorescence polarization to have a reduction in binding affinity to TbtF⁵. By creating an alanine positional scanning library, we demonstrated that our method could achieve similar results to those using fluorescence polarization as evidenced by a greater than 100-fold decrease in AlphaLISA signal compared to the wild-type peptide sequence for all noted mutations.”

Figure 2: Why are the authors including “SRRGS” at the beginning of the TbtA peptide? This portion is clearly a leftover from a cloning or cleavage step. Of course, mutation of these

residues to Ala will have no contribution, they are not part of the native peptide! TbtA starts with Met. I am worried that the authors are using these residues improperly in other % identity calculations. Stating that the peptide 38% TbtA sequence also seems to be misleading to me; isn't this peptide 17/24 the same (71% TbtA)?

We apologize for the confusion. We initially included the "SRRGS" linker in our construct as this linker had been used previously by Burkhart et al. when performing fluorescence polarization assays with TbtA⁶. For consistency, as well as to serve as a linker between the N-terminal sFLAG tag recognized by the AlphaLISA beads and the TbtA peptide sequence, we chose not to remove the sequence in our downstream assays. However, to reflect that the "SRRGS" would not be expected to influence binding between TbtA and TbtF, we have now removed that portion of the sequence from our analysis and schematics.

For the % identity calculations mentioned above, the values provided in **Figure 2** are determined by calculating what percentage of the synthetic peptide match the leader sequence of TbtA. Each synthetic peptide was composed of 40 amino acids, but for simplicity we only depicted residues between the -34 and -17 position of the leader sequence because that was the region identified as being most important for binding recognition in our alanine scan (Figure 2a). This explains why our % identity calculations are lower than may be calculated if considering only the residues depicted in the figure. To clarify this point, we have added the following text to the figure caption on p. 6:

"For simplicity, only amino acids between the -34 and -17 position are depicted, however, each peptide was composed of 40 amino acids, reflecting the length of the TbtA leader sequence with an additional 5 amino acid linker."

Line 225: This statement about RREs being incorrectly identified as a challenge for RiPP discovery is not true. There is a reliable, HHpred-based tool for RRE detection that three groups developed a number of years back, Medema, van Wezel, Mitchell, which is now quite popular. Also, as the authors point out, many are readily detected as homologs of PqqD which is a simple BLAST search. I have never heard of a single case of a misidentified RRE being a problem in RiPP discovery.

Thank you for pointing out that the reliability of RiPP prediction tools has vastly improved in the last several years. We have removed our statement on the misidentification via these tools as a major problem in the RiPP discovery pipeline.

Line 227: The authors here claim they are working on an uncharacterized lasso peptide predicted by AntiSMASH. As a matter of fact, this lasso peptide was bioinformatically reported in 2017 (Nat Chem Biol, Mitchell) and it was experimentally characterized in 2023 (referred to as LP-1010) (Nat Microbiol, Voigt).

Thank you for bringing to our attention these reports on the lasso peptide (Las-1010) that we also produced and characterized in our work. It was not our intent to leave these important reports absent from our manuscript. We have revised our manuscript to include both and to contextualize our results considering these reports. We have recentered the focus of the manuscript to highlight our method and have removed language about the 'uncharacterized' nature of Las-1010. Below we outline several changes we made to incorporate these reports.

In terms of computational identification, when we generated our dataset of predicted lasso peptide BGCs, we included a deduplication step to remove any predicted lasso peptide

sequences that had already been reported in literature. However, we only removed those sequences that included experimental data, with the rationale that our workflow could still be used to validate computational predictions. At the time we carried out our analysis, the 2023 paper was not published. We have incorporated additional text on p. 8 to include clarifying details on how we curated our set of clusters:

“Sequences that matched computationally predicted but not experimentally verified sequences reported in the literature were maintained in the dataset while those that had been characterized experimentally at the time of our analysis were removed; in doing so, we reasoned that our workflow could help validate predictions generated by others in the field.”

In terms of Las-1010 specifically, we certainly did not intend to misrepresent our characterization of Las-1010 nor leave the previously published work on this peptide uncited. Between the time that our work was experimentally completed, and our manuscript was submitted, experimental characterization of Las-1010 was reported in the 2023 publication. Despite not being the first to publish characterization of this lasso peptide, we were able to use our screen for down-selecting predicted BGCs, leading to our production of Las-1010 independent of the work by King et al⁷. *In vitro* workflows like ours that prototype computationally predicted RiPP BGCs⁸ can inform which computationally predicted RiPP products are worth pursuing *in vivo* through this down-selection process. Nevertheless, it was still our responsibility to survey the newly published literature before our submission, and we regrettably missed this important reference. To correct this and enhance our manuscript, we have amended the text in several locations to incorporate the King et al. publication and reframe the purpose of our work around screening and down-selection rather than novelty of a particular lasso peptide.

On p. 11,

“During our work, King et al. reported for the first time the heterologous production of this lasso peptide (termed Las-1010) in E. coli from the same biosynthetic cluster⁷. Las-1010, which was previously bioinformatically identified⁹ but not experimentally characterized, was found by King et al. to exhibit weak antibacterial activity against some bacterial strains⁷.”

On p. 7,

“Successful heterologous expression of computationally predicted RiPP products in vivo can be a challenge due to the inability to precisely control expression timing and yield, as well as the absence of necessary cofactors¹⁰.”

On p. 18-19,

“Additionally, by coupling our workflow with computational prediction tools, we demonstrated how our platform can screen for natural product BGCs likely to function in an in vitro setting. While we were able to produce Las-1010⁷ (Las24), many of the clusters we detected binding activity for in our screen did not produce mature lasso peptides. We hypothesize that this could be due to a number of reasons, including that the in vitro reaction environment may lack other important components natively included in in vivo systems, such as auxiliary genes as has been demonstrated for other natural products^{11,12}. Advances in developing cell-free lysates from non-E. coli-based organisms, such as Streptomyces¹³⁻¹⁶, could be incorporated into future studies as a method for testing expression systems that contain native auxiliary factors. Despite this, by prioritizing BGCs with a demonstrated functional first step (RRE binding), our methods can be

used as is to narrow down the number of proteins needed to be expressed and purified for attempts at in vitro reconstitution. Furthermore, we note that recent advances in deep learning models^{17,18} and protein language models¹⁹ have accelerated our ability to predict the substrate promiscuity of RiPP biosynthetic enzymes. Key to these computational tools is the ability to rapidly generate training datasets and validate any resulting predictions. Our workflow can be interfaced with these computational tools.”

More broadly, we now also expand the application of our screening workflow to characterize multiple types of post-translational modifications, such as glycosylation. This is shown in new **Figures 4-6**. Our hope is this adds additional emphasis on the power of the method rather than the previously described novelty of the lasso peptide.

Line 238: It is odd that the authors use antiSMASH to construct a much smaller dataset than those published by other groups. At minimum, it should be acknowledged that their dataset is not “a complete collection”, as it is considerably smaller than other published datasets.

We apologize for the confusion regarding our statement. The reason our dataset is smaller than other published collections is that we included a filtering step to only include clusters that contained the complete consortium of annotated biosynthetic genes required for lasso peptide biosynthesis. To clarify this point, we have edited the noted sentence and incorporated additional text into the methods section.

Main text edit on p. 8

*“Of these, 1,882 BGCs were predicted to contain all essential lasso peptide biosynthetic enzymes (**Supplementary Table 3**).”*

Addition to methods section:

*“A total of 2,574 lasso peptide clusters were identified, and from this set we then performed an additional filtering step to identify 1,882 BGCs which contained a complete collection of essential biosynthetic enzymes (**Supplementary Table 3**). Specifically, we note that predictions using AntiSMASH rely on identifying clusters in which the predicted components include homology to PF13471 and a proximal asparagine synthetase, micJ25, or mcJC. Therefore, there is a possibility that clusters identified by AntiSMASH are missing essential enzymes (which we did observe) and we did not include these incomplete clusters in our follow-up analysis.”*

Line 297: As stated above, Alphasassin has previously been characterized by the Voigt group (referred to as LP-1010). Does this peptide have antibacterial activity? How much was isolated? Why stress the cationic features and not do these experiments?

Because our initial goal was to identify lasso peptides with antimicrobial activity, we included a BGC prioritization step that included characteristics related to potential antibacterial activity. We did not perform any additional antibacterial activity assays. However, it turns out that Las24 (initially discovered as Las-1010) was tested for antimicrobial activity. King et al. found it to exhibit weak antibacterial activity against some bacterial strains. While we did not discover any additional lasso peptides with antimicrobial activity, we felt that our filtering criterion was an important context to provide as it played a role in which BGCs were selected for prototyping and which were not. We added additional text on p. 11 noting that King et al. tested that Las-1010 for antibacterial activity⁷.

“Las-1010, which was previously bioinformatically identified⁹ but not experimentally characterized, was found by King et al. to exhibit weak antibacterial activity against some bacterial strains⁷.”

MINOR ISSUES:

Although most journals do not require it, it is advisable if all authors would include accession identifiers for all proteins discussed in the main text of their manuscripts. Biochemistry requires this and Springer Nature should too. This helps literature mining algorithms that link protein sequences with experimentally determined functional data.

Thank you for this suggestion, we have incorporated the noted accession numbers into our extended data set where we have also included the sequences of all proteins used in this study.

In the RiPP field, it is customary to use negative numbering for the leader peptide. The authors should follow the convention established by the RiPP community. And please do not include residues that are not part of the native peptide.

We have updated all figure and text referring to specific leader sequence residues with the appropriate negative numbering scheme. We have also removed the non-native portions of the peptide and updated our calculations in **Figure 2** appropriately.

Line 66: PqqD is not an “enzyme”. It is a PqqA-binding domain. The enzyme formally is PqqE. See work from Klinman and Latham.

Thank you for correcting us. While reframing the manuscript to include additional applications, we have removed this sentence altogether.

Line 121: For the sake of clarity, the term PqqD should only be used when referring to bona fide PqqD. When referring to an RRE from some other RiPP pathway, “PqqD” should be explicitly avoided. Use RRE or the name of the protein, if it has one.

We agree and have edited the noted instance on p. 4 of PqqD to RRE.

“For 9 of these proteins, we tested the native sequence as well as fusion proteins in which the predicted RRE domain was fused to maltose-binding protein (MBP) due to their size and/or origin from a radical S-adenosyl-L-methionine (SAM) enzyme that could potentially make expression difficult.”

Line 150: odd to call this pathway a derivative of GE2270, when the pathway encodes thiomuracin, a known thiopeptide with many supporting publications.

As noted above, we have amended the text on p. 5 to refer to this pathway as the thiomuracin pathway.

Lines 155-170: the AlphaLISA method appears strong regarding throughput, but it appears to only be qualitative and somewhat lacking in detecting small differences in binding strengths. The authors seem to minimize the effectiveness of other assays (like ITC, NMR or FP), despite their lower throughput nature. One must admit the data gleaned from these older methods is more quantitative and sensitivity to binding changes is readily detected. I don't see that as being the case for AlphaLISA. Perhaps these two assays will become complementary, with AlphaLISA

used as a first-pass rather than it supplanting other methods, as the narrative currently implies. It is important to balance the pros and cons of any new method, not just sell the pros.

Thank you for this point; we agree that providing a more nuanced discussion of the benefits and drawbacks of our method is important context to provide. We have incorporated additional text within the discussion to address this concern.

“One important limitation to our work is that the results we obtain are semi-quantitative. With our current platform design, we can compare the relative binding affinity of different RRE-peptide pairs or glycosylation efficiency of specific OST mutants but are unable to provide exact quantitative measurements of these phenomena (e.g., k_d , % glycosylation, etc.). Thus, we suggest that our complementary method can be integrated with more traditional assays by first using our workflow as a screening tool to down-select specific protein variants from larger libraries for follow-up experiments with smaller numbers of samples.”

Lines 199-200. This is the mean of 3 technical replicates, but what is the standard deviation? Do we get a K_d out of these data? How reproducible are these data upon biological replication? Prep-to-prep variability is often an issue, for instance. For establishment of a new method, this is critical data to include.

Thank you for bringing up these important questions regarding variability and reproducibility. To provide the reader with a sense of variability with the AlphaLISA assay, we have included multiple Supplementary Figures in which we compare the results of biological replicate reactions. In **Supplementary Figure 7**, we test all computationally predicted lasso peptide RRE-precursor peptide pairs in biological triplicate and observe low variability as evidenced by the error bars indicating standard deviation included with each condition. Additionally, **Supplementary Figures 20 and 24** show parity plots for biological replicates fitted to the line $y=x$ with calculated R^2 values. These plots demonstrate a strong agreement between replicates, with R^2 values between ~ 0.93 and 0.96 .

With respect to determining a K_d , we are not able to determine a K_d with the presented experimental set-up. However, a K_d could be determined using a competition binding assay, as has been done previously^{20,21}. We do note, however, that while our method as presented cannot provide precise binding kinetic values, our results for the TbtA alanine scan correspond closely with published IC_{50} data, where larger increases in IC_{50} result in a larger decrease in AlphaLISA signal. In particular, the below paragraph from our results section discusses these findings:

*“We next asked whether we could characterize a peptide-binding landscape to inform the design of a synthetic peptide capable of binding to a naturally occurring RRE. To do this, we chose the RRE domain of TbtF, the cyclodehydratase involved in thiomuracin^{4,5} biosynthesis, and its leader sequence of TbtA (**Fig. 2a**). Mutating residues L(-32), L(-29), M(-27), D(-26), and F(-24) within the leader sequence of TbtA to an alanine was previously shown using fluorescence polarization to have a reduction in binding affinity to TbtF⁵. By creating an alanine positional scanning library, we demonstrate that our method could achieve similar results to those using fluorescence polarization as evidenced by a greater than 100-fold decrease in AlphaLISA signal compared to the wild-type peptide sequence for all noted mutations. We also found that the mutation D(-30)A resulted in a greater than 100-fold decrease in AlphaLISA signal. By using CFE combined with AlphaLISA, we were able to characterize the peptide-binding landscape within hours without conventional cloning, transformation, expression, and purification workflows normally required for fluorescence polarization competition assays.”*

Lines 245 and 247: the term “pro-peptide” is not encouraged, see 2013 Arnison et al community review on RiPP guidelines and nomenclature.

Thank you for this suggestion. We have edited all instances of “pro-peptide” to the appropriate terms based on the supplied reference.

Line 248: This seems like a bait and switch. Are the authors trying to find lasso peptides that act as an antibiotic? If so, where are the antibacterial assays?

Our intent was to be upfront about this, and we have modified the text on this as mentioned above with the last “Major Issue”. Specifically, we mention that we do not identify any new antibacterial activity but prioritized BGCs that included characteristics related to potential antibacterial activity.

Figure 3: Which examples had multiple RREs and how was this determined? Did RRE-Finder confirm these predictions? The authors should use the best tools available, not older insufficient tools that make their results sound more appealing. Many of these data are from a single replicate, thus without replication should not be published. Others have five replicates. I am confused by the lack of consistency and lack of rational.

Using AntiSMASH 5.1.2, we were able to find intriguing clusters where multiple precursor peptides and enzymes were encoded in a cluster or in overlapping clusters, as shown in **Figure 3**. By looking at the gene context, it was difficult to confidently assign enzymes to specific precursor peptides or precursor peptides to enzymes when multiple were present. All identified BGCs with multiple RREs are presented in **Supplementary Figure 6**. After we completed our bioinformatic search, AntiSMASH 6.0²² was released, which incorporated RRE-Finder as you state.

With respect to the number of replicates, our intention with **Figure 3** is to demonstrate how our workflow can be used as a screening step, with further validation experiments confirming the results of our screen. Additionally, as we note in the main text, we then repeated every RRE-peptide pair with an independent experiment in biological triplicate (**Supplementary Figure 7**), which confirmed the results of our initial screen. Thus, our results are reproducible.

*“A subsequent validation experiment assaying all RRE-peptide pairs in biological triplicate at the dilution condition that yielded the highest AlphaLISA signal confirmed the results of our screen, with RRE-peptide pairs that produced higher AlphaLISA signal in our initial screen generally producing higher AlphaLISA signal in the validation experiment (**Supplementary Fig. 7**).”*

Line 311: a MALDI instrument typically cannot give this much accuracy for a mass measurement. Do not report non-significant figures.

We have updated the work with the appropriate number of significant figures.

Line 316: Notwithstanding the former comment, a MALDI instrument should not be off by 1 Da either. Something is wrong here. The error is most likely in the tenths place (not the ones place nor the ten-thousands place, per last comment).

Thank you for catching this oversight; we returned to the original MALDI spectra and the correct m/z that we observed is 2360.2. We have updated the text to reflect this point.

Lines 360 to 362: I have never heard of a case where the misannotation of RiPP biosynthetic enzymes was the offending issue. Poor expression/solubility of the biosynthetic enzymes in a heterologous host, however, is a notorious problem in RiPP biosynthesis and should be considered here.

We agree that poor expression and solubility of RiPP tailoring enzymes in heterologous hosts is often an issue in reconstituting RiPP BGCs. However, for many of the lasso peptide BGCs that we tested, we were able to obtain soluble protein through both cell-free expression and *E. coli in vivo* heterologous expression and were still unable to produce a mature product. We hypothesize that this could be due to other differences between native and *in vitro* settings, such as the absence of other gene products not identified in the bioinformatic BGC prediction (which we point out on p. 18). We have removed the sentence to avoid any misrepresentations.

Lines 366-368: This statement is an oversimplification and problematic for several reasons: Just because an RRE is active does not mean the rest of the enzymes will be active. As shown by Klinman, Nair, Naismith, Severinov, van der Donk, Mitchell, Piel, Hegemann, Schulman, and a great number of additional other labs, RREs are easy to express and purify in functional form. Other RiPP biosynthetic components are typically much less well behaved and thus it is misleading to state the claim the authors are making. Conversely, just because an RRE failed to express in the PurExpress setting (which lacks chaperones, crowding effects, and the general robustness of a living cell) does not mean it would not work in a heterologous expression system. There is a large amount of literature to support the opposite of what the authors state here. Lastly, I am uncertain of how the 40% figure was calculated.

We apologize for the confusion regarding the noted sentence. We agree that even in instances where an active and soluble RRE is readily expressed, other components of the BGC may not be. Indeed, many of the clusters that we observed functional RRE-peptide interactions for in **Figure 2**, we were unable to completely reconstitute to produce the mature lasso peptide. Our intent was only to point out that our workflow can help down-select which clusters are most suitable for *in vitro* reconstitution, not definitively determine which clusters will ultimately produce a mature lasso peptide.

We have amended the noted sentence on p. 18 to the following:

“Despite this, by prioritizing BGCs with a demonstrated functional first step (RRE binding), our methods can be used as is to narrow down the number of proteins needed to be expressed and purified for attempts at in vitro reconstitution.”

With respect to potential unsuccessful expression in PURExpress, we have incorporated **Supplementary Figure 8** which demonstrates soluble expression of all computationally predicted lasso peptide RREs, suggesting unsuccessful expression is not the major challenge in our workflow. With the change in manuscript framing, we also removed references to the 40% figure you mention.

The SI would be a good place to enhance rigor and data transparency for the lasso peptide section. Showing Coomassie-stained SDS-PAGE gels as well for every protein used in the study is a must. Accession codes, or at the very least the amino acid sequence, should be given for all of the lasso peptide precursor peptides.

We agree that providing protein gels for all enzymes we purified and tested for activity is important. We have updated the supplementary figures to include additional protein gels. **Supplementary Figure 8** demonstrates soluble expression in PURE_{flex} of all computationally predicted lasso peptide RREs tested in this study. **Supplementary Figure 11** shows a Coomassie SDS-PAGE gel of purified tailoring enzymes used to reconstitute the Las24 (Las-1010) BGC. **Supplementary Figure 17** shows a Coomassie SDS-PAGE gel of purified tailoring enzymes from the Fuscanodin/Fusilassin BGC. We have also provided the sequences for all lasso peptide precursor and tailoring enzyme genes in the extended dataset.

Reviewer #3 (Remarks to the Author):

In this manuscript, Wong et al describe a system that has the potential to improve the speed and scale of RRE/precursor peptide identification. Ribosomal natural products (RiPPs) are produced through the expression of a precursor peptide that gets post-translationally modified by one or more enzymes. Most RiPPs discovered to date possess a leader peptide, which is important for recognition by parts of the biosynthetic machinery. In about 50% of RiPP biosynthetic gene clusters one finds one or more RiPP recognition element(s) (RREs), which are responsible for binding to the leader peptide. The presented study uses a cell-free expression system to produce RRE/precursor peptide pairs and an in-solution ELISA to probe the interaction. I found the manuscript very well written and the method presented exciting. In my mind, the study is going to attract a lot of attention amongst RiPP researchers and provides an interesting starting point for other fields. I therefore recommend publication in Nat Commun after the following issues have been addressed:

Thank you for celebrating the broad interest of our method and recommending publication.

1. The authors express the RREs as MBP fusion proteins, presumably because the RREs cannot be expressed without this solubility tag. This results in two issues:
 - a) The authors should include a negative control using just MBP, without an RRE, to demonstrate that the interaction is specific.
 - b) The authors should characterize several of the RREs once they have been cleaved off MBP. It is a well known problem that MBP is powerful enough to keep misfolded and aggregated proteins in solution, and the expression in Supplementary Figure 1 does not look very strong, another indicator that the RRE may be "unhappy".

Thank you for the suggestion of these experiments; we agree that demonstration of these points confirms the accuracy and utility of our approach. We have incorporated **Supplementary Figure 2** and additional conditions in **Supplementary Figure 7**, which demonstrates that MBP alone does not bind to any of the tested precursor peptides. We have also incorporated **Supplementary Figure 3** which demonstrates that we can express RREs in PURE_{flex} without fusion to MBP and still retain binding activity to their associated precursor peptide.

2. The authors characterize a large number of RRE/precursor peptide pairs from lasso peptide clusters identified via genome mining. While in principle a great way of probing many interactions in parallel, I could not find data demonstrating expression of all the RRE-MBP fusions employed. Are the non-binders actually not binding, or the RRE simply not expressed? The authors should show at least a gel for the RRE-MBP fusions of figure 3.

We agree that demonstrating successful expression of each of the screened RREs is an important validation step and have incorporated **Supplementary Figure 8** which uses

Fluorotect™ to demonstrate successful expression of each RRE screened in **Figure 3**.

3. The authors claim to have made a new lasso peptide. To make that claim requires the authors to isolate that products, or at least detect it, in the natural producer. Otherwise the wording should be amended to assert the production of a peptide with the topology of a lasso peptide. Which brings me to the analytics of the new lasso peptide: While treatment with carboxypeptidase Y is a start, more experiments are required to assert the proposed structure/topology is correct. Don't worry, I understand that an NMR would be unreasonable, but MS fragmentation data could easily be acquired with the small quantities produced.

We agree with you that our initial characterization experiments were incomplete. To further bolster our conclusions, we scaled-up reactions producing Las24 (Las-1010) and performed LC-MS/MS to confirm the identity and structure of our product. We have provided the spectra from this experiment in **Supplementary Figure 14** and have referenced this figure in the main text of the manuscript on p.11.

*“Subsequent characterization experiments confirmed that the production of this lasso peptide, Las24, is time dependent (**Supplementary Fig. 12**), that each of the proteins in the predicted BGC is necessary for maturation (**Supplementary Fig. 13**), that the sequence of the molecule matches the expected structure (**Supplementary Fig. 14**), that the molecule is resistant to carboxypeptidase (a common confirmation of threaded topology) (**Supplementary Fig. 15**), and that there is limited to no interaction of Las24 biosynthetic components with those from other lasso peptide clusters (**Supplementary Fig. 16**). ”*

With respect to our phrasing of producing a lasso peptide, we have updated the text on p. 11 with the suggested phrasing:

“...we successfully produced one peptide with the topology of a lasso peptide, from BGC 24”

References

- 1 Koos, J. D. & Link, A. J. Heterologous and in Vitro Reconstitution of Fuscanodin, a Lasso Peptide from *Thermobifida fusca*. *Journal of the American Chemical Society* **141**, 928-935 (2019). <https://doi.org/10.1021/jacs.8b10724>
- 2 DiCaprio, A. J., Firouzbakht, A., Hudson, G. A. & Mitchell, D. A. Enzymatic Reconstitution and Biosynthetic Investigation of the Lasso Peptide Fusilassin. *J Am Chem Soc* **141**, 290-297 (2019). <https://doi.org/10.1021/jacs.8b09928>
- 3 Kloosterman Alexander, M., Shelton Kyle, E., van Wezel Gilles, P., Medema Marnix, H. & Mitchell Douglas, A. RRE-Finder: a Genome-Mining Tool for Class-Independent RiPP Discovery. *mSystems* **5**, e00267-00220 (2020). <https://doi.org/10.1128/mSystems.00267-20>
- 4 Hudson, G. A., Zhang, Z., Tietz, J. I., Mitchell, D. A. & van der Donk, W. A. In Vitro Biosynthesis of the Core Scaffold of the Thiopeptide Thiomuracin. *J Am Chem Soc* **137**, 16012-16015 (2015). <https://doi.org/10.1021/jacs.5b10194>
- 5 Zhang, Z. *et al.* Biosynthetic Timing and Substrate Specificity for the Thiopeptide Thiomuracin. *Journal of the American Chemical Society* **138**, 15511-15514 (2016). <https://doi.org/10.1021/jacs.6b08987>
- 6 Burkhart, B. J., Hudson, G. A., Dunbar, K. L. & Mitchell, D. A. A prevalent peptide-binding domain guides ribosomal natural product biosynthesis. *Nature Chemical Biology* **11**, 564-570 (2015). <https://doi.org/10.1038/nchembio.1856>

- 7 King, A. M. *et al.* Systematic mining of the human microbiome identifies antimicrobial peptides with diverse activity spectra. *Nature Microbiology* **8**, 2420-2434 (2023). <https://doi.org/10.1038/s41564-023-01524-6>
- 8 Montalbán-López, M. *et al.* New developments in RiPP discovery, enzymology and engineering. *Natural Product Reports* **38**, 130-239 (2021). <https://doi.org/10.1039/D0NP00027B>
- 9 Tietz, J. I. *et al.* A new genome-mining tool redefines the lasso peptide biosynthetic landscape. *Nature chemical biology* **13**, 470-478 (2017). <https://doi.org/10.1038/nchembio.2319>
- 10 Montalbán-López, M. *et al.* New developments in RiPP discovery, enzymology and engineering. *Nat Prod Rep* **38**, 130-239 (2021). <https://doi.org/10.1039/d0np00027b>
- 11 Garcie, C. *et al.* The Bacterial Stress-Responsive Hsp90 Chaperone (HtpG) Is Required for the Production of the Genotoxin Colibactin and the Siderophore Yersiniabactin in *Escherichia coli*. *The Journal of Infectious Diseases* **214**, 916-924 (2016). <https://doi.org/10.1093/infdis/jiw294>
- 12 Washio, K., Lim, S. P., Roongsawang, N. & Morikawa, M. Identification and Characterization of the Genes Responsible for the Production of the Cyclic Lipopeptide Arthrofactin by *Pseudomonas* sp. MIS38. *Bioscience, Biotechnology, and Biochemistry* **74**, 992-999 (2010). <https://doi.org/10.1271/bbb.90860>
- 13 Xu, H., Liu, W. Q. & Li, J. A *Streptomyces*-Based Cell-Free Protein Synthesis System for High-Level Protein Expression. *Methods Mol Biol* **2433**, 89-103 (2022). https://doi.org/10.1007/978-1-0716-1998-8_5
- 14 Des Soye, B. J., Davidson, S. R., Weinstock, M. T., Gibson, D. G. & Jewett, M. C. Establishing a High-Yielding Cell-Free Protein Synthesis Platform Derived from *Vibrio natriegens*. *ACS Synthetic Biology* **7**, 2245-2255 (2018). <https://doi.org/10.1021/acssynbio.8b00252>
- 15 Li, J., Wang, H. & Jewett, M. C. Expanding the palette of *Streptomyces*-based cell-free protein synthesis systems with enhanced yields. *Biochemical Engineering Journal* **130**, 29-33 (2018). <https://doi.org/https://doi.org/10.1016/j.bej.2017.11.013>
- 16 Moore, S. J. *et al.* A *Streptomyces venezuelae* Cell-Free Toolkit for Synthetic Biology. *ACS Synthetic Biology* **10**, 402-411 (2021). <https://doi.org/10.1021/acssynbio.0c00581>
- 17 Vinogradov, A. A., Chang, J. S., Onaka, H., Goto, Y. & Suga, H. Accurate Models of Substrate Preferences of Post-Translational Modification Enzymes from a Combination of mRNA Display and Deep Learning. *ACS Central Science* **8**, 814-824 (2022). <https://doi.org/10.1021/acscentsci.2c00223>
- 18 Chang, J. S., Vinogradov, A. A., Zhang, Y., Goto, Y. & Suga, H. Deep Learning-Driven Library Design for the De Novo Discovery of Bioactive Thiopeptides. *ACS Central Science* **9**, 2150-2160 (2023). <https://doi.org/10.1021/acscentsci.3c00957>
- 19 Mi, X., Barrett, S. E., Mitchell, D. A. & Shukla, D. LassoESM: A tailored language model for enhanced lasso peptide property prediction. *bioRxiv*, 2024.2010.2025.620295 (2024). <https://doi.org/10.1101/2024.10.25.620295>
- 20 Lazar, G. A. *et al.* Engineered antibody Fc variants with enhanced effector function. *Proc Natl Acad Sci U S A* **103**, 4005-4010 (2006). <https://doi.org/10.1073/pnas.0508123103>
- 21 Hunt, A. C. *et al.* Multivalent designed proteins neutralize SARS-CoV-2 variants of concern and confer protection against infection in mice. *Science Translational Medicine* **14**, eabn1252 (2022). <https://doi.org/doi:10.1126/scitranslmed.abn1252>
- 22 Blin, K. *et al.* antiSMASH 6.0: improving cluster detection and comparison capabilities. *Nucleic Acids Res* **49**, W29-W35 (2021). <https://doi.org/10.1093/nar/gkab335>

Reviewer #1 (Remarks to the Author):

The authors have addressed most all of the concerns raised during the initial review of this manuscript. I would suggest addition of a few more references (especially with regards to work on the thiopeptide system) for the sake of completeness. Otherwise, this is fine for publication.

We thank you for recommending publication. We have added several new thiopeptide and RiPP references. We now also specifically call out the term “thiopeptide” here (p. 4):

“We chose to first apply our workflow to RiPPs (e.g., lanthipeptides^{1,2}, thiopeptides³⁻⁵) due to growing interest in their use as antimicrobial therapeutics⁶⁻¹².”

The new references include:

- Schwalen, C.J., Hudson, G.A., Kille, B., & Mitchell, D.A. Bioinformatic expansion and discovery of thiopeptide antibiotics. *J Am Chem Soc* **140**, 9494-9501 (2019). <https://doi.org/10.1021/jacs.8b03896>
- Pelton, J.M. et al. Cheminformatics-Guided Cell-Free Exploration of Peptide Natural Products. *J Am Chem Soc* **146**, 8016-8030 (2024). <https://doi.org/10.1021/jacs.3c11306>
- Rice, A.J. et al. Enzymatic Pyridine Aromatization during Thiopeptide Biosynthesis. *J Am Chem Soc* **144**, 21116-21124 (2022). <https://doi.org/10.1021/jacs.2c07377>
- Vinogradov, A.A. et al. De Novo Discovery of Thiopeptide Pseudo-natural Products Acting as Potent and Selective TNIK Kinase Inhibitors. *J Am Chem Soc* **144**, (2022). <https://doi.org/10.1021/jacs.2c07937>
- Rice, A.J. et al. Cell-free synthetic biology for natural product biosynthesis and discovery. *Chem. Soc. Rev.* (2025). [10.1039/d4cs01198h](https://doi.org/10.1039/d4cs01198h)
- Zhang, Z.J., et al. Activity of Gut-Derived Nisin-like Lantibiotics against Human Gut Pathogens and Commensals. *ACS Chem. Biol.* **19**, 357-369 (2024). <https://doi.org/10.1021/acscchembio.3c00577>
- Repka, L.M., Chekan, J.R., Nair, S.K., & van der Donk, W.A. Mechanistic Understanding of Lanthipeptide Biosynthetic Enzymes. *Chem Rev.* **117**, 5457-5520 (2017). <https://doi.org/10.1021/acs.chemrev.6b00591>

With respect to the work on the thiomuracin pathway, we have also incorporated additional references into the following sentence (p. 6):

“To do this, we chose the RRE domain of TbtF, the cyclodehydratase involved in thiomuracin¹³⁻¹⁶ biosynthesis, and its leader sequence of TbtA (Fig. 2a).”

The new references include:

- Fleming, S.R. et al. Flexizyme-enabled benchtop biosynthesis of thiopeptides. *J Am Chem Soc* **141**, 758-762 (2019). [10.1021/jacs.8b11521](https://doi.org/10.1021/jacs.8b11521)
- Cogan, D.P., et al. Structural insights into enzymatic [4+2] aza-cycloaddition in thiopeptide antibiotic biosynthesis. *Proc. Natl. Acad. Sci* **114**, 12928-12933 (2017). <https://doi.org/10.1073/pnas.1716035114>

Reviewer #2 (Remarks to the Author):

The authors did a tremendous job in responding to the prior critique. The current manuscript removed all of the concerning issues and even added a new dimension of data with glycosylation. I believe the manuscript is ready for publication.

We thank you for celebrating the work that went into our revision and for recommending publication.

Reviewer #4 (Remarks to the Author):

In the present manuscript Wong and colleagues describe their design of a cell-free expression system, which is high throughput and adaptable for screening large libraries of carrier protein and OST mutants. Moreover, the system is adaptable to study posttranslational modifications, including protein glycosylation. As my expertise is on glycobiology, I have focused my review on this aspect of the work.

We thank you for highlighting the potential impact of our high-throughput methods for studying post-translational modifications.

The authors make quite strong claims on their workflow being suitable for vaccine design. Although I agree that their workflow is suitable for accelerated design and perhaps high throughput screening of vaccines, in my opinion the actual proof of their concept is lacking. For vaccines to work the product needs to be as pure as possible. Yet, the authors generate a variety of products as can be seen in the western blots of supplementary figures 18 and 21. The authors claim these are longer CPS fragments, but as they used a “crude membrane fraction” these products could indeed be longer CPS variants, but the addition of completely other or multiple polysaccharide fragment is a real possibility. The anti-CPS4 anti-serum used for instance in figure 4, proves that CPS4 is present, but doesn’t prove that no other products are also included in their mixture. Therefore, to fully verify their claims the authors should include a mass spectrometry glycoproteomic analysis, including at least the different products they get in the glycoprotein sfGFP and CPS4 reaction.

We thank you for raising these points. Your comments helped us realize that we need to: (1) better focus our claims on the goal of the study, which was to develop a robust, high-throughput, and generalizable workflow that expedites the ability to characterize and engineer PTMs on peptides and proteins, rather than vaccine production/design, (2) highlight that others have previously shown production of the CPS4 glycan using the plasmids we used in the study, and (3) clarify the nature of our crude membrane fraction only being able to conjugate carrier proteins with the CPS4 glycan in our system, as opposed to other glycans. Below, we describe how we have addressed these concerns.

1. We removed reference to vaccine design, which was not our focus, and now better emphasize our technology application to two classes of post-translational modifications (PTMs). For example, rather than state that we produced conjugate vaccines, the abstract is now focused on production of glycosylated proteins using a model vaccine carrier protein.

In the abstract, we write (p. 2):

“We apply our workflow to two representative classes of peptide and protein therapeutics: ribosomally synthesized and post-translationally modified peptides (RiPPs) and glycoproteins.”

Additionally, in the abstract we write (p. 2):

“Then, we adapt our workflow to study and engineer oligosaccharyltransferases (OSTs) involved in protein glycan coupling technology, leading to the identification of mutant OSTs and sites within a model vaccine carrier protein that enable high efficiency production of glycosylated proteins.”

In the introduction, we write (p. 3):

“Here, we describe a general in vitro, plate-based platform for characterizing and engineering PTMs using CFE and AlphaLISA which we apply to both (i) ribosomally synthesized and post-translationally modified peptides (RiPPs) and (ii) glycoproteins.”

To reduce the emphasis on vaccine development, we removed the following statement on p.16: “...to form *S. pneumoniae* serotype 4 conjugate vaccines, which could allow for careful engineering of conjugate expression, glycosylation efficiency, and possibly even immunogenicity.” The text now reads:

“We sought to discover which sites within a clinically relevant carrier protein can be efficiently, enzymatically glycosylated in vitro.”

We also incorporated the word “model” in front of the phrase “vaccine carrier protein” on p.15 to emphasize that our goal was to demonstrate our workflow with a potential carrier protein, rather than produce conjugate vaccine products for use in downstream immune studies. The text now reads (p. 15):

“With an efficient OST in hand, we next asked at which locations throughout a model vaccine carrier protein we could attach the bacterial polysaccharide.”

In the discussion, we note that our emphasis is on developing efficient glycosylation systems, which will help us engineer systems for vaccine production, rather than having made vaccines themselves (p. 18):

“...towards engineering systems for efficient conjugate vaccine production, including protein engineering to increase the efficiency of a glycan-installing enzyme.”

In the discussion, we also removed the phrase “Using conjugate vaccines as a test system...” to focus the purpose of our work on engineering enzymes and substrates used in glycoprotein synthesis systems. We similarly replaced the term “conjugate vaccines” with “glycoproteins.” The new text now reads (p. 19):

*“We also demonstrated how our workflow can be used to engineer both enzymes and substrates used in glycoprotein synthesis systems. With our platform, we rapidly assessed 285 unique mutants of CjPglB to enable efficient production of glycoproteins with the CPS from *S. pneumoniae* serotype 4, a major cause of pneumonia in disadvantaged communities¹⁷.”*

Together, these changes more aptly place the focus on the workflow for optimizing post-translational modifications for increased glycosylation efficiency rather than vaccine design. That said, we have previously created protective, immunogenic vaccines using this approach targeting *Francisella tularensis* (Stark et al., *Science Advances*, 2021) and enterotoxigenic *E. coli* (Warfel et al., *ACS Synthetic Biology*, 2022; Williams et al., *Frontiers in Molecular Biosciences*, 2023), which both have a similar distribution of polysaccharide lengths. Thus, we believe our approach could be useful for making conjugate vaccines in the future.

2. Regarding the identity of glycosylation observed on Western blot, previous work in bacterial glycoengineering validates our approach. For example, all previous efforts to our knowledge have shown that the desired O-antigens or capsular polysaccharides produced using recombinant biosynthetic pathways in *E. coli* are attached to proteins in varying chain lengths, producing a ladder pattern when visualized on Western blot¹⁸⁻²⁴, and multiple studies have shown that the correct sequence of the polysaccharide repeat unit is observed when analyzed by mass spectrometry^{19,20}. With respect to the CPS4 polysaccharide, the plasmid we used to synthesize CPS4 has been described previously and was confirmed to produce CPS4 antigen²³. Additionally, CPS4 glycan synthesized in *E. coli* using this same plasmid has been used to synthesize conjugate vaccines, which were shown to produce antibodies that target *S. pneumoniae* serotype 4²⁵. To improve clarity regarding our system, we have added the following text to the manuscript (p. 12):

“The CPS4 glycan is composed of the repeating tetrasaccharide unit PyrGal-ManNAc-FucNAc-GalNAc (PyrGal: pyruvate attached to galactose; ManNAc: N-acetylmannosamine; FucNAc: N-acetylfucosamine; GalNAc: N-acetylgalactosamine) and is important for conjugate vaccine protection against pneumococcal infection²³. Other groups have previously shown that the CPS4 glycan can be synthesized in strains of E. coli via recombinant expression of the CPS4 biosynthetic pathway and attached to proteins using the OST PglB from Campylobacter jejuni (CjPglB)²³⁻²⁵.”

3. There also appears to be confusion over which CPS glycans could be attached to the carrier protein based on our use of “crude membrane extracts.” To be clear, only CPS4 could be attached as it is the only glycan being overexpressed. We have added the following text to page 12:

“The crude membrane fraction is produced from E. coli cells expressing a single biosynthetic pathway encoding a single pathogen-specific O-antigen or capsular polysaccharide.”

We also added additional description of the method we used to isolate CMF (p. 12):

“We began by overexpressing the CPS4 glycan in E. coli cells, harvesting and lysing the cells, and concentrating the membrane vesicles containing CPS4 via ultracentrifugation to produce CPS4-enriched crude membrane fractions.”

Since this protein-glycan coupling approach, as well as the specific plasmid encoding for the biosynthetic pathway for CPS4 biosynthesis, have been extensively validated in previous publications, we believe our use of an anti-CPS4 antibody combined with the observed characteristic banding pattern above the aglycosylated protein on Western blots is sufficient confirmation of our glycoprotein product.

That said, we have also carried out two additional experiments.

First, we now show a dot blot that confirms the presence of CPS4 in our crude membrane fraction (CMF) as new **Supplementary Figures 20 and 21**. We added the following text to the manuscript to describe this result (p. 12):

*“Following verification of the presence of the CPS4 glycan in our CMF with an anti-CPS4 dot blot (**Supplementary Fig. 20**), we then showed...”*

Second, we performed an anti-6xHis and an anti-CPS4 blot on duplicate *in vitro* glycosylation reactions prepared with both blank CMF and CPS4 CMF. Previously, we did not include an anti-CPS4 Western blot. These blots prove that our carrier protein only gets glycosylated with CPS4 glycan when we use CPS4 enriched CMF. Furthermore, our carrier protein does not get glycosylated when blank CMF is added to the reaction, confirming there are no glycans other than CPS4 being transferred to the carrier protein. On the same blots, we also confirm that the CPS4 glycan is transferred to the carrier protein when using two top PglB mutants identified in our mutagenesis screen. These additional experiments verify the identity of the CPS4 glycan and highlight that our method indeed enables high-efficiency enzymatic decoration of the carrier protein with the CPS4 glycan. These blots are shown as new **Supplementary Figures 22 and 23**.

To describe these results, we added the following text to the manuscript (p. 12):

*“IVG reactions using CMF prepared from cells without CPS4 overexpression confirmed that only CPS4 is being transferred onto the carrier protein in our system, while an anti-CPS4 Western blot confirmed identity of the CPS4 glycan on our glycoconjugates (**Supplementary Fig. 22**).”*

We also added the following (p. 14):

"An anti-CPS4 Western blot confirmed transfer of the CPS4 glycan using two top PglB hits (Supplementary Fig. 22)."

Moreover, also the non-glycosylated AQNAT has an additional product, which seems enriched in the DQNAT. What is this additional band?

The additional band seen in the AQNAT control lanes (**Supp. Fig. 18b**) is likely due to background signal from the anti-6xHis primary antibody, which is known to bind to other proteins in a sample. This conclusion is supported by the absence of this band in the negative control lane in **Supplementary Fig. 25**, which uses an anti-myc primary antibody.

References:

- 1 Zhang, Z. J. *et al.* Activity of Gut-Derived Nisin-like Lantibiotics against Human Gut Pathogens and Commensals. *ACS Chemical Biology* **19**, 357-369 (2024). <https://doi.org/10.1021/acscchembio.3c00577>
- 2 Repka, L. M., Chekan, J. R., Nair, S. K. & van der Donk, W. A. Mechanistic Understanding of Lanthipeptide Biosynthetic Enzymes. *Chemical Reviews* **117**, 5457-5520 (2017). <https://doi.org/10.1021/acs.chemrev.6b00591>
- 3 Schwalen, C. J., Hudson, G. A., Kille, B. & Mitchell, D. A. Bioinformatic Expansion and Discovery of Thiopeptide Antibiotics. *J Am Chem Soc* **140**, 9494-9501 (2018). <https://doi.org/10.1021/jacs.8b03896>
- 4 Rice, A. J. *et al.* Enzymatic Pyridine Aromatization during Thiopeptide Biosynthesis. *Journal of the American Chemical Society* **144**, 21116-21124 (2022). <https://doi.org/10.1021/jacs.2c07377>
- 5 Vinogradov, A. A. *et al.* De Novo Discovery of Thiopeptide Pseudo-natural Products Acting as Potent and Selective TNIK Kinase Inhibitors. *Journal of the American Chemical Society* **144**, 20332-20341 (2022). <https://doi.org/10.1021/jacs.2c07937>
- 6 Ayikpoe, R. S. *et al.* A scalable platform to discover antimicrobials of ribosomal origin. *Nature Communications* **13**, 6135 (2022). <https://doi.org/10.1038/s41467-022-33890-w>
- 7 Hudson, G. A. & Mitchell, D. A. RiPP antibiotics: biosynthesis and engineering potential. *Current Opinion in Microbiology* **45**, 61-69 (2018). <https://doi.org/https://doi.org/10.1016/j.mib.2018.02.010>
- 8 Fu, Y., Jaarsma, A. H. & Kuipers, O. P. Antiviral activities and applications of ribosomally synthesized and post-translationally modified peptides (RiPPs). *Cellular and Molecular Life Sciences* **78**, 3921-3940 (2021). <https://doi.org/10.1007/s00018-021-03759-0>
- 9 Shin, J. M. *et al.* Biomedical applications of nisin. *J Appl Microbiol* **120**, 1449-1465 (2016). <https://doi.org/10.1111/jam.13033>
- 10 Chiumento, S. *et al.* Ruminococcin C, a promising antibiotic produced by a human gut symbiont. *Science Advances* **5**, eaaw9969 (2019). <https://doi.org/doi:10.1126/sciadv.aaw9969>
- 11 Pelton, J. M. *et al.* Cheminformatics-Guided Cell-Free Exploration of Peptide Natural Products. *Journal of the American Chemical Society* **146**, 8016-8030 (2024). <https://doi.org/10.1021/jacs.3c11306>
- 12 Rice, A. J. *et al.* Cell-free synthetic biology for natural product biosynthesis and discovery. *Chem Soc Rev* (2025). <https://doi.org/10.1039/d4cs01198h>
- 13 Hudson, G. A., Zhang, Z., Tietz, J. I., Mitchell, D. A. & van der Donk, W. A. In Vitro Biosynthesis of the Core Scaffold of the Thiopeptide Thiomuracin. *J Am Chem Soc* **137**, 16012-16015 (2015). <https://doi.org/10.1021/jacs.5b10194>
- 14 Zhang, Z. *et al.* Biosynthetic Timing and Substrate Specificity for the Thiopeptide Thiomuracin. *Journal of the American Chemical Society* **138**, 15511-15514 (2016). <https://doi.org/10.1021/jacs.6b08987>
- 15 Fleming, S. R. *et al.* Flexizyme-Enabled Benchtop Biosynthesis of Thiopeptides. *J Am Chem Soc* **141**, 758-762 (2019). <https://doi.org/10.1021/jacs.8b11521>
- 16 Cogan, D. P. *et al.* Structural insights into enzymatic [4+2] aza-cycloaddition in thiopeptide antibiotic biosynthesis. *Proceedings of the National Academy of Sciences* **114**, 12928-12933 (2017). <https://doi.org/doi:10.1073/pnas.1716035114>

- 17 Kobayashi, M. L., Andrew J.; Gierke, Ryan; Farrar, Jennifer L.; Morgan, Rebecca L.; Campos-Outcalt, Doug; Schechter, Robert; Poehling, Katherine A.; Long, Sarah S.; Loehr, Jamie; Cohen, Adam L. Use of 21-Valent Pneumococcal Conjugate Vaccine Among U.S. Adults: Recommendations of the Advisory Committee on Immunization Practices - United States, 2024. *Morbidity and Mortality Weekly Report* **73**, 793-798 (2024). <https://doi.org/http://dx.doi.org/10.15585/mmwr.mm7336a3>
- 18 Wacker, M. *et al.* Substrate specificity of bacterial oligosaccharyltransferase suggests a common transfer mechanism for the bacterial and eukaryotic systems. *Proceedings of the National Academy of Sciences* **103**, 7088-7093 (2006). <https://doi.org/doi:10.1073/pnas.0509207103>
- 19 Feldman, M. F. *et al.* Engineering N-linked protein glycosylation with diverse O antigen lipopolysaccharide structures in *Escherichia coli*. *Proc Natl Acad Sci U S A* **102**, 3016-3021 (2005). <https://doi.org/10.1073/pnas.0500044102>
- 20 Knoot, C. J. *et al.* Discovery and characterization of a new class of O-linking oligosaccharyltransferases from the Moraxellaceae family. *Glycobiology* **33**, 57-74 (2022). <https://doi.org/10.1093/glycob/cwac070>
- 21 Warfel, K. F. *et al.* A Low-Cost, Thermally Stable, Cell-Free Protein Synthesis Platform for On-Demand Production of Conjugate Vaccines. *ACS Synthetic Biology* **12**, 95-107 (2023). <https://doi.org/10.1021/acssynbio.2c00392>
- 22 Williams AJ, W. K., Desai P, Li J, Lee J, Wong DA, Nguyen PM, Qin Y, Sobol SE, Jewett MC, Chang Y, DeLisa MP. A low-cost recombinant glycoconjugate vaccine confers immunogenicity and protection against enterotoxigenic *Escherichia coli* infections in mice. *Frontiers in Molecular Biosciences* **10** (2023). <https://doi.org/10.3389/fmolb.2023.1085887>
- 23 Kay, E. J., Yates, L. E., Terra, V. S., Cuccui, J. & Wren, B. W. Recombinant expression of *Streptococcus pneumoniae* capsular polysaccharides in *Escherichia coli*. *Open Biol* **6**, 150243 (2016). <https://doi.org/10.1098/rsob.150243>
- 24 Kay, E. J. *et al.* Engineering a suite of *E. coli* strains for enhanced expression of bacterial polysaccharides and glycoconjugate vaccines. *Microb Cell Fact* **21**, 66 (2022). <https://doi.org/10.1186/s12934-022-01792-7>
- 25 Reglinski, M. *et al.* A recombinant conjugated pneumococcal vaccine that protects against murine infections with a similar efficacy to Prevnar-13. *NPJ Vaccines* **3**, 53 (2018). <https://doi.org/10.1038/s41541-018-0090-4>